# Mechanistic and evolutionary insights into isoform-specific 'supercharging' in DCLK family kinases

Aarya Venkat[1†], Grace Watterson[1†], Dominic P Byrne[2†], Brady O'Boyle[1], Safal Shrestha[3], Nathan Gravel[3], Emma E Fairweather[2], Leonard A Daly[2,4], Claire Bunn[1], Wayland Yeung[3], Ishan Aggarwal[1], Samiksha Katiyar[1], Claire E Eyers[2,4], Patrick A Eyers[2]*, Natarajan Kannan[3]*

[1]Department of Biochemistry and Molecular Biology, University of Georgia, Athens, United States; [2]Department of Biochemistry, Cell and Systems Biology, Institute of Systems, Molecular and Integrative Biology, University of Liverpool, Liverpool, United Kingdom; [3]Institute of Bioinformatics, University of Georgia, Athens, United States; [4]Centre for Proteome Research, Department of Biochemistry, Cell and Systems Biology, Institute of Systems, Molecular and Integrative Biology, University of Liverpool, Liverpool, United Kingdom

*For correspondence:
paeyers@liverpool.ac.uk (PAE);
nkannan@uga.edu (NK)

†These authors contributed equally to this work

Competing interest: The authors declare that no competing interests exist.

**Abstract** Catalytic signaling outputs of protein kinases are dynamically regulated by an array of structural mechanisms, including allosteric interactions mediated by intrinsically disordered segments flanking the conserved catalytic domain. The doublecortin-like kinases (DCLKs) are a family of microtubule-associated proteins characterized by a flexible C-terminal autoregulatory 'tail' segment that varies in length across the various human DCLK isoforms. However, the mechanism whereby these isoform-specific variations contribute to unique modes of autoregulation is not well understood. Here, we employ a combination of statistical sequence analysis, molecular dynamics simulations, and in vitro mutational analysis to define hallmarks of DCLK family evolutionary divergence, including analysis of splice variants within the DCLK1 sub-family, which arise through alternative codon usage and serve to 'supercharge' the inhibitory potential of the DCLK1 C-tail. We identify co-conserved motifs that readily distinguish DCLKs from all other calcium calmodulin kinases (CAMKs), and a 'Swiss Army' assembly of distinct motifs that tether the C-terminal tail to conserved ATP and substrate-binding regions of the catalytic domain to generate a scaffold for autoregulation through C-tail dynamics. Consistently, deletions and mutations that alter C-terminal tail length or interfere with co-conserved interactions within the catalytic domain alter intrinsic protein stability, nucleotide/inhibitor binding, and catalytic activity, suggesting isoform-specific regulation of activity through alternative splicing. Our studies provide a detailed framework for investigating kinome-wide regulation of catalytic output through cis-regulatory events mediated by intrinsically disordered segments, opening new avenues for the design of mechanistically divergent DCLK1 modulators, stabilizers, or degraders.

## eLife assessment

This **important** study expands on current knowledge of allosteric diversity in the human kinome by C-terminal splicing variants using as a paradigm DCLK1. The authors provide **convincing** evolutionary and some mechanistic evidence how C-terminal isoform specific variants generated by alternative splicing can regulate catalytic activity by means of coupling specific phosphorylation sites to dynamical and conformational changes controlling active site and substrate pocket occupancy,

as well as protein-protein interactions. The data will be of interest to researchers in the kinase and signal transduction field.

## Introduction

Protein kinases are one of the largest druggable protein families comprising 1.7% of the human genome and play essential roles in regulating diverse eukaryotic cell signaling pathways (*Manning et al., 2002*). The doublecortin-like kinases (DCLKs) are understudied members of the calcium/calmodulin-dependent kinase (CAMK) clade of serine-threonine kinases (*Agulto et al., 2021*; *Bayer and Schulman, 2019*; *Gógl et al., 2019*). There are three distinct paralogs of DCLK (1, 2, and 3), the last of which is annotated as a 'dark' kinase due to the lack of information pertaining to its function (*Berginski et al., 2021*). Human DCLK1 (also known as DCAMKL1) was initially identified in 1999 (*Sossey-Alaoui and Srivastava, 1999*), followed by the cloning of human DCLK2 and -3 paralogs (*Ohmae et al., 2006*). Full-length DCLK proteins contain N-terminal doublecortin-like (DCX) domains, microtubule-binding elements that play a role in microtubule dynamics, neurogenesis, and neuronal migration (*Couillard-Despres et al., 2005*; *Horesh et al., 1999*). DCLKs have garnered much interest as disease biomarkers, since they are upregulated in a variety of cancer pathologies (*Cheng et al., 2022a*; *Gao et al., 2016*; *Westphalen et al., 2017*), as well as neurodegenerative disorders such as Huntington's disease (*Galvan et al., 2018*). However, the mechanisms by which DCLK activity is autoregulated, and how and why they have diverged from other protein kinases is not well understood.

Like all protein kinases, the catalytic domain of DCLKs adopts a bi-lobal fold (*Gógl et al., 2019*), with an N-terminal ATP-binding lobe and C-terminal substrate-binding region. Canonical elements within the two lobes include the DFG motif, a Lys-Glu salt bridge that is associated with the active conformation, Gly-rich loop, and ATP-binding pocket, which are all critical elements for catalysis. Many protein kinases, including CAMKs, tyrosine kinases, and AGCs, elaborate on these core elements with unique N-terminal and C-terminal extensions that flank these catalytic lobes (*Kannan et al., 2007*; *Yeon et al., 2016*; *Nguyen et al., 2015a*; *Yeung et al., 2021*; *Kwon et al., 2018*), allowing them to function as allosteric regulators of catalytic activity (*Gógl et al., 2019*). Indeed, CAMKs are archetypal examples of kinases that can exist in an active-like structural conformation, yet still remain catalytically inactive (*Gógl et al., 2019*). This is in large part due to the presence of unique C-terminal tails that are capable of blocking ATP or substrate binding in well-studied kinases such as CAMK1 and CAMKII. In canonical CAMKs, autoinhibition may be released upon $Ca^{2+}$/calmodulin (CaM) interaction with the CAMK C-tail, which makes the substrate-binding pocket and enzyme active site accessible (*Rellos et al., 2010*). In CAMKII, the N- and C-terminal segments flanking the kinase domain are variable in length across different isoforms and the level of kinase autoinhibition or autoactivation has been reported to be dependent on the linker length (*Bhattacharyya et al., 2020*). The CAMKII C-tail can be organized into an autoregulatory domain and an intrinsically disordered association domain. The autoregulatory domain also serves as a pseudosubstrate, which physically blocks the substrate-binding pocket until it is competed away by CaM (*Hudmon and Schulman, 2002*). Notably, this autoregulatory pseudosubstrate can be phosphorylated (*Rellos et al., 2010*), and phosphorylation of the C-tail makes CAMKII insensitive to CaM binding. Across the CAMK group, several other kinases share autoinhibitory activity via interactions between $Ca^{2+}$/CaM binding domains and the C-terminal tail (*Huse and Kuriyan, 2002*; *Wayman et al., 2008*), and a major feature of these kinases is variation in the tail length across the distinct genetic isoforms.

The human genome encodes four distinct DCLK1 isoforms, termed DCLK1.1–1.4 in UniProt (*Table 1*, *Figure 1*; *Omori et al., 1998*), which display differential activity- and tissue-specific expression profiles. Human DCLK1.1 (also known as DCLK1 alpha) is expressed in a variety of

**Table 1.** Names of doublecortin-like kinase (DCLK) isoforms discussed in this paper, along with their respective isoform number, UniProt identification, and alternate names that have been used.

| Name (this Study) | Isoform number | UniProt ID | Alternate names |
|---|---|---|---|
| DCLK1.1 | 1 | O15075-2 | DCAMKL1 alpha |
| DCLK1.2 | 2 | O15075-1 | DCAMKL1 beta |
| ΔDCLK1.1 | 3 | O15075-3 | |
| ΔDCLK1.2 | 4 | O15075-4 | |

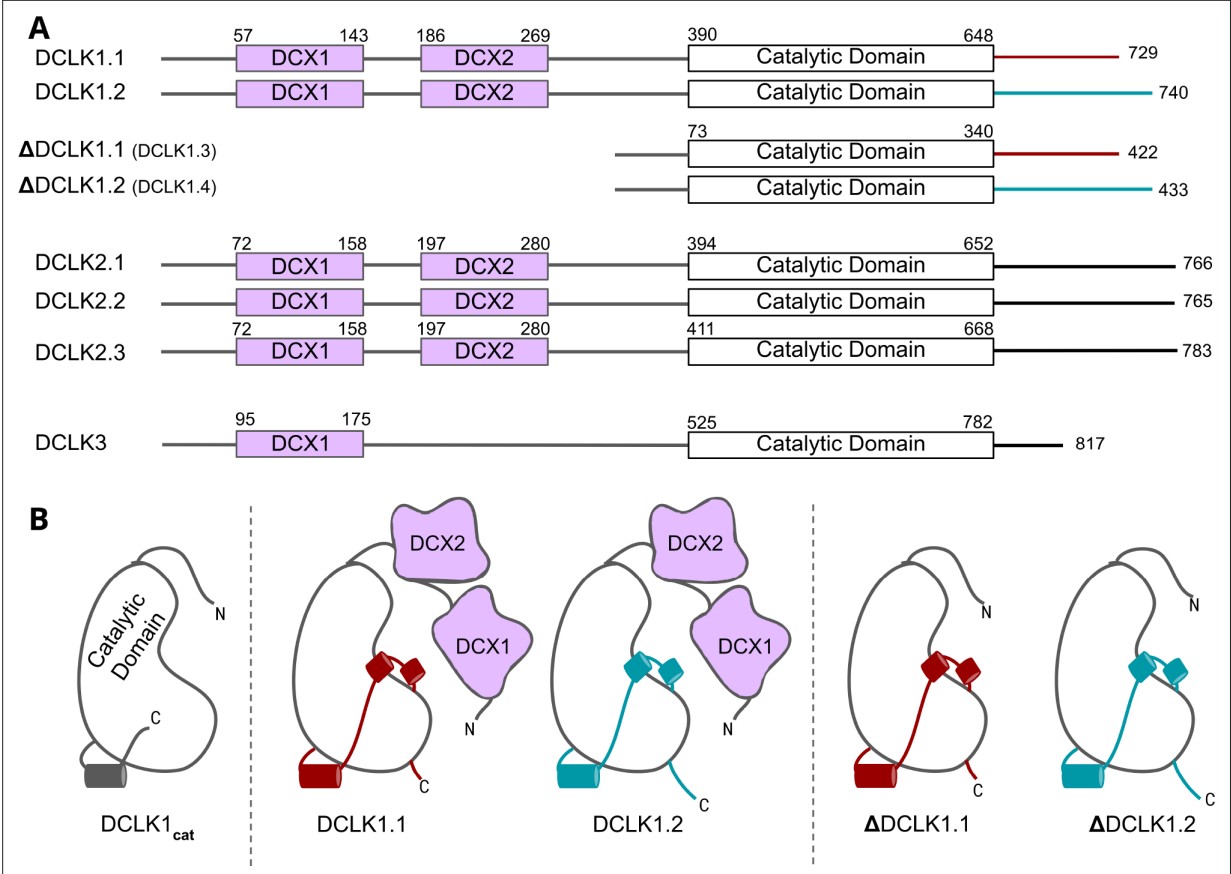

**Figure 1.** Cartoon schematic of Human DCLK orthologs and respective isoforms. (**A**) Schematic representation of domain organization for the known isoforms of the three human doublecortin-like kinase (DCLK) paralogs. Domain boundaries are annotated according to the representative amino acid sequences derived from UniProt. (**B**) DCLK1 isoforms visualized as cartoons, showing key structural differences between the four human DCLK1 isoforms and a DCLK1 catalytic domain with artificially short linker regions (DCLK1cat).

The online version of this article includes the following figure supplement(s) for figure 1:

**Figure supplement 1.** Structural cartoon depicting each DCLK1 isoform, categorized by the presence of doublecortin-like (DCX) domain and related to enzymatic activity.

tissues, but is enriched in cells derived from the fetal and adult brain, whereas DCLK1.2 (also known as DCLK1 beta) is expressed exclusively in the embryonic brain (*Matsumoto et al., 1999*). DCLK1.3 and DCLK1.4, which lack tandem microtubule-binding DCX domains (*Figure 1*) but are otherwise identical to DCLK1.1 and DCLK1.2, respectively, are also highly expressed in the brain. To aid with clarity, the names of the human DCLK1 genes and their isoforms used in this paper are summarized in *Table 1*. Recent structural and cellular analyses have begun to clarify the mechanisms by which the DCLK1.2 isoform is regulated by the C-tail (*Agulto et al., 2021*; *Patel et al., 2021*; *Cheng et al., 2022b*). Mechanistically, autophosphorylation of Thr 688, which is present only in the C-tail of DCLK1.2 (and DCLK1.4), blunts kinase activity and subsequently inhibits phosphorylation of the N-terminal DCX domain and thus drives DCLK microtubule association in cells (*Agulto et al., 2021*). Consistently, deletion of the C-tail or mutation of Thr 688 restores DCLK1.2 kinase activity, subsequently leading to DCX domain phosphorylation and the abolition of microtubule binding. The length and sequence of the C-tail varies across the DCLK1 isoforms; however, how these variations contribute to isoform-specific functions and how they emerged during the course of evolution is not known.

In this paper, we employ an evolutionary systems approach that combines statistical sequence analysis with experimental studies to generate new models of DCLK evolutionary divergence and functional specialization. We identify the C-terminal tail as the hallmark of DCLK functional specialization across the kingdoms of life and propose a refined model in which this regulated tail functions as a highly adaptable 'Swiss Army Knife' that can 'supercharge' multiple aspects of DCLK signaling

output. Notably, a conserved segment of the C-tail functions as an isoform-specific autoinhibitory motif, which mimics ATP functions through direct tail docking to the nucleotide-binding pocket, where it forms an ordered set of interactions that aligns the catalytic (C) spine of the kinase in the absence of ATP binding. Furthermore, molecular modeling demonstrates that a phosphorylated threonine in the C-tail of DCLK1.2, which is absent in DCLK1.1, is positionally poised to competitively mimic the gamma phosphate of ATP, perhaps in a regulated manner. Other segments of the tail function as a pseudosubstrate by occluding the substrate-binding pocket and tethering to key functional regions of the catalytic domain. Thermostability analysis of purified DCLK1 proteins, combined with molecular dynamics (MD) simulations, confirms major differences in thermal and dynamic profiles of the DCLK1 isoforms, while catalytic activity assays reveal how specific variations in the G-loop and C-tail can rescue DCLK1.2 from the autoinhibited conformation. Together, these studies demonstrate that isoform-specific variations in the C-terminal tail co-evolved with residues in the DCLK kinase domain, contributing to regulatory diversification and functional specialization.

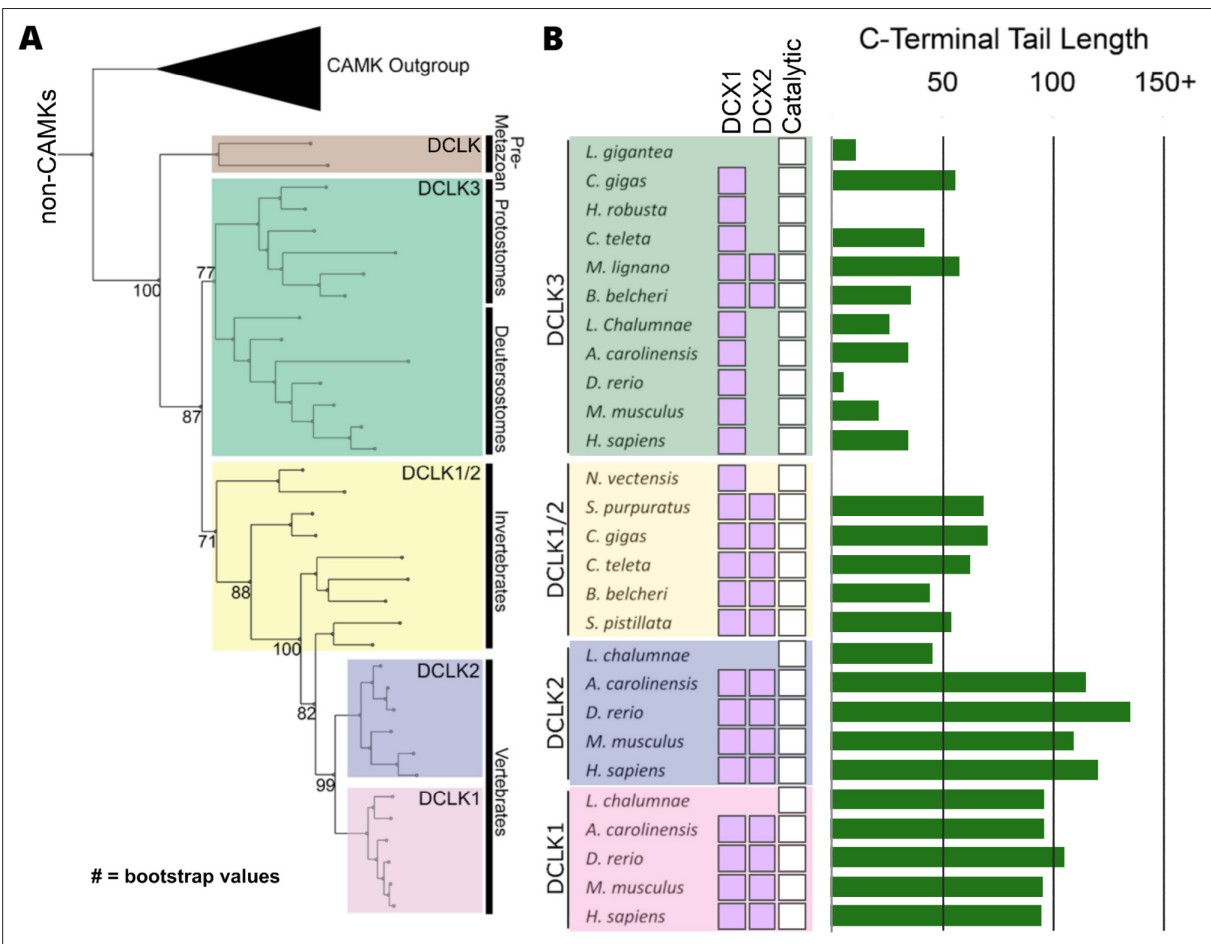

**Figure 2.** Evolution of the doublecortin-like kinase (DCLK) family. (**A**) Phylogenetic tree showing the divergence and grouping of DCLK sub-families in different taxonomic groups. Bootstrap values are provided for each clade. (**B**) shows domain annotations for sequences included in the phylogenetic tree. The length of C-terminal tail segment for these sequences is shown as a histogram (green). The original tree generated using IQTREE is provided in *Figure 2—source data 1*.

The online version of this article includes the following source data and figure supplement(s) for figure 2:

**Source data 1.** Full doublecortin-like kinase (DCLK) phylogenetic tree with bootstrap values.

**Source data 2.** Species and Uniprot IDs for each sequence represented in the domain plot of *Figure 2*.

**Figure supplement 1.** Electrostatic surface views of full-length doublecortin-like kinase (DCLK) paralog (**A**: DCLK1, **B**: DCLK2, and **C**: DCLK3) in the same orientation.

# Results

## Origin and evolutionary divergence of DCLK family members

The human DCLKs' repertoire is composed of three genes, termed DCLK1, -2, and -3 (**Figure 1A**, **Table 1**). The experimental model employed in this study, DCLK1, is composed of multiple spliced variants in human cells. Those full-length proteins that contain N-terminal DCX domains are usually referred to as DCLK1.1 or DCLK1.2 and the variants that lack the DCX domains are termed here (for simplicity) ΔDCLK 1.1 and ΔDCLK1.2 (also referred to as DCLK1.3 and DCLK1.4). The core catalytic domain with minimal flanking regions (DCLK1$_{cat}$, **Figure 1B**) is identical in all DCLK1 proteins, whereas the length of the tail, or the presence of the DCX domains, generates considerable diversity from the single human DCLK1 gene (**Figure 1B**, **Figure 1—figure supplement 1**). To infer evolutionary relationships of DCLK paralogs, and especially the evolution of the C-terminal tail regions that lie adjacent to the kinase domain (**Figure 1**), we performed phylogenetic analysis of 36 DCLK sequences with an outgroup of closely related CAMK sequences (**Figure 2A**, **Figure 2—source data 1**). These DCLK sequences are from a representative group of holozoans, which consist of multicellular eukaryotes (metazoans) and closely related unicellular eukaryotes (pre-metazoans). The analysis generated four distinct clades: pre-metazoan DCLK, metazoan DCLK3, vertebrate DCLK2, and vertebrate DCLK1. Interestingly, DCLK genes demonstrated significant expansion and diversification within metazoan taxa. The pre-metazoan DCLK sequences were the most ancestral and showed no DCLK diversity, suggesting the DCLK expansion and diversification correlated with the evolution of multicellular organisms. Within the metazoan expansion of DCLK, DCLK3 is the most ancestral and can be broken down into two sub-clades: protostome DCLK3 and deuterostome DCLK3. Within invertebrates, only two DCLK paralogs were present, one that was identified as a DCLK3 ortholog and another that was not clearly defined as either DCLK1 or DCLK2. This suggests that the diversification into DCLK1 and DCLK2 paralogs from an ancestral DCLK1/2-like paralog occurred after the divergence of invertebrates and vertebrates, which is further supported by the monophyletic DCLK1 and DCLK2 clades in vertebrates (bootstrap value: 99).

Interestingly, the expansion of DCLK in metazoans and the diversification of DCLK1 and DCLK2 within vertebrates correlates well with the length and sequence similarity of the C-terminal tail, which also varies between the different DCLK1 splice variants (**Figure 1A**, **Figure 1—figure supplement 1** and **Figure 2—figure supplement 1**). Within both protostome and deuterostome DCLK3, the length of the C-terminal tail is ~50 residues or less. This is in marked contrast to the tail lengths of vertebrate DCLK1 and DCLK2, which are ~100 residues long. In addition to the C-terminal tail, an analysis of the domain organization of these DCLKs reveals that DCLK3 predominantly contains only a single N-terminal doublecortin domain (DCX), whereas invertebrate DCLKs, and vertebrate DCLK1 and DCLK2 predominantly contain two DCX domains at the N-terminus of the long isoforms (**Figure 2B**). In addition, we identified a putative active site-binding motif, VSVI, and a phosphorylatable threonine conserved within vertebrate DCLK1 and DCLK2, which is absent in all other DCLK sequences, including invertebrate DCLK1/2. This raises the possibility that the DCLK1/2 tail extensions are employed for vertebrate-specific regulatory functions.

Next, we compared the type of DCLK1 protein sequence encoded by a range of chordate mammalian genomes. The domain organization of each DCLK1 isoform was compared based on annotated sequences from UniProt, demonstrating the presence of at least one DCLK1 protein that lacks the DCX domains in every species examined, with a mixture of ΔDCLK1.1 and ΔDCLK1.2 splice variants. Interestingly, it was only in the human DCLK1 gene that definitive evidence for ΔDCLK1.1 and ΔDCLK1.2 variants were found (**Figure 3A**). To establish a model for DCLK1 biophysical analysis, we constructed a recombinant hybrid human DCLK1 catalytic domain with a short C-tail sequence that is equivalent to DCLK1.1 amino acids 351–689, containing the catalytic domain with a short C-tail region. As shown in **Figure 3B**, incubation of size exclusion chromatography (SEC)-purified GST-tagged DCLK1 with 3C protease generated the mature untagged DCLK1 protein for biophysical analysis. Analytical SEC revealed that purified DCLK1.1 and DCLK1.2 isoforms are monomeric in solution (**Figure 3—figure supplements 1–3**). We evaluated catalytic activity for DCLK1.1$_{351-689}$ using a validated peptide phosphorylation assay (**Figure 3C**), which revealed efficient phosphorylation of a DCLK1 substrate peptide. The $K_{M[ATP]}$ for peptide phosphorylation was close to 20 µM in the presence of Mg$^{2+}$ ions (**Figure 3C**, left panel), similar to values measured for other Ser/Thr kinases that are autophosphorylated and active after expression from bacteria (**Byrne et al., 2020b**). DCLK-dependent peptide phosphorylation was

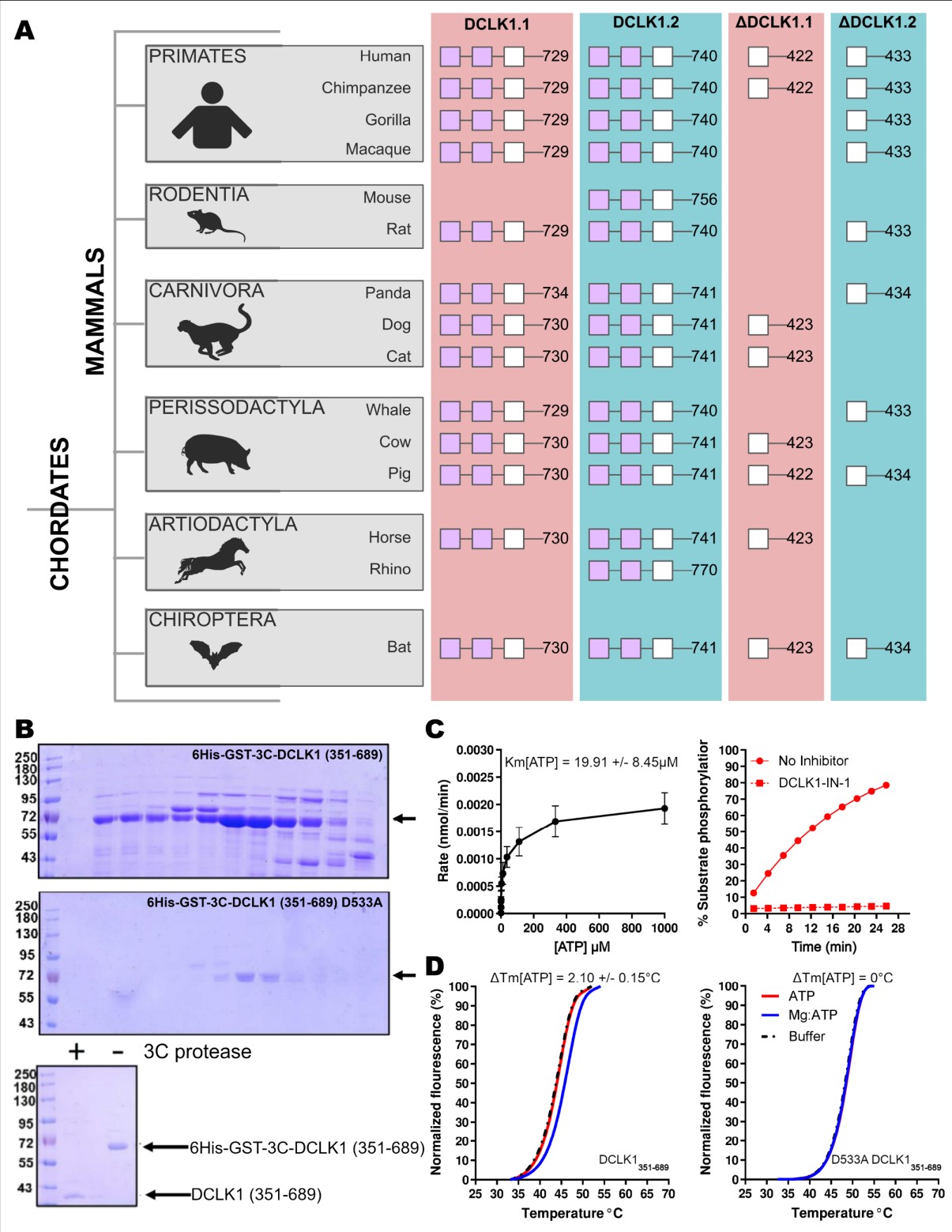

**Figure 3.** Cartoon cladogram of mammalian DCLK1 isoforms and experimental. (**A**) Cartoon cladogram of mammalian species showing the domain organization of each DCLK1 isoform from representative annotated sequences from UniProt. UniProt IDs for each sequence are provided in *Figure 3—source data 1*. (**B**) SDS-PAGE of 6His-GST-3C-DCLK1.1 (351–689, top) or a D533A mutant in which the DFG Asp is mutated to Ala (middle). Proteins were separated by size exclusion chromatography, and high-purity fractions were pooled. The affinity tag was removed prior to analysis by incubation with 3C protease, leading to a demonstrable shift in mobility (bottom). (**C**) Evaluation of catalytic activity toward DCLK1 peptide. DCLK1.1 351–689

*Figure 3 continued on next page*

*Figure 3 continued*

possesses a $K_{M[ATP]}$~20 µM in vitro (left) and real-time substrate phosphorylation was inhibited by prior incubation with the small molecule DCLK1-IN-1 (right). (**D**) Thermal shift assay demonstrating a 2.1°C increase in the stability of DCLK1 351–689 in the presence of Mg:ATP (left), which was absent in the D533A protein (right). Raw data are provided in *Figure 3—source data 2*.

The online version of this article includes the following source data and figure supplement(s) for figure 3:

**Source data 1.** Species and Uniprot IDs for each sequence represented in the mammalian DCLK1 isoform species tree.

**Source data 2.** Experimental values for thermal shift and activity assays from *Figure 3*.

**Source data 3.** DCLK1 plasmid sequences generated for *Figure 3*.

**Source data 4.** *Figure 3B* (top) uncropped gel.

**Source data 5.** *Figure 3B* (bottom) uncropped gel.

**Source data 6.** Uncropped gel of incubation of size exclusion chromatography (SEC)-purified DCLK1 with 3C protease.

**Source data 7.** PowerPoint file containing labels for Source Data 4–6.

**Figure supplement 1.** Characterization and Analysis of Human DCLK1 Variants and Their Biophysical Properties.

**Figure supplement 1—source data 1.** Uncropped and unlabeled gel from DCLK1.1 size exclusion chromatography (SEC).

**Figure supplement 2.** Analysis and Confirmation of Monomeric DCLK1.2$^{351-740}$ Through Size Exclusion Chromatography and SDS-PAGE.

**Figure supplement 2—source data 1.** Uncropped and unlabeled gel from DCLK1.2 size exclusion chromatography (SEC).

**Figure supplement 3.** Analytical size exclusion chromatography (SEC) of DCLK1.1$^{351-729}$ and DCLK1.2$^{351-740}$.

**Figure supplement 4.** Differential scanning fluorimetry (DSF) profile of DCLK1$_{351-689}$ in the presence of DMSO or a panel of DCLK1 inhibitor compounds.

completely blocked (*Figure 3C*, right panel) by prior incubation of the reaction mixture (containing 1 mM ATP) with the chemical inhibitor DCLK1-IN-1, as expected (*Ferguson et al., 2020*). In addition to enzyme activity, we monitored thermal denaturation of purified, folded, DCLK1$_{351-689}$ protein in the presence of ATP, either alone or as a Mg:ATP complex, which is required for catalysis. As shown in *Figure 3D*, DCLK1 was stabilized by 2.1°C upon incubation with an excess of Mg:ATP, and this protective effect was completely blocked by mutation of Asp 533 (of the conserved DFG motif) to Ala, consistent with canonical ATP interaction in the nucleotide-binding site. Finally, we assessed the thermal effects of a panel of DCLK1 inhibitors on the model DCLK1.1$_{351-689}$ protein. Prior incubation with DCLK1-IN-1, LRRK2-IN-1, the benzopyrimidodiazipinones XMD8-92 and XMD8-85, which have been reported to potently (though not specifically) inhibit DCLK1 activity (*Patel et al., 2021*), led to marked protection from thermal unfolding (*Figure 3—figure supplement 4*). Consistently, the negative control compound DCLK1-Neg (*Ferguson et al., 2020*) was ineffective in stabilizing DCLK1.

## Key differences between isoforms in the C-tail of DCLK1 arise from alternative-splicing and different open-reading frames

Higher-order vertebrates have multiple isoforms of DCLK1 and DCLK2, where sequence variations occur in either or both the N- and C-terminal regions attached to the kinase domain. Human DCLK1, for example, has four unique isoforms. Isoforms 1 and 2 differ in C-terminal tail length due to variations in exon splicing (*Figure 4A*). Further examination of the intron and exon boundaries indicates that human DCLK1.1 contains an additional exon (exon 16) that is not spliced in DCLK1.2. Exon 16 is spliced with exon 17 with a phase 2 intron, which introduces a shift in the reading frame and an earlier translated stop codon (UGA) in exon 17 (*Figure 4B*). In DCLK1.2, exon 15 is spliced with exon 17, with a non-disruptive phase 0 intron, resulting in the full translation of exon 17. These changes introduce multiple indels (insertions and deletions) and result in the insertion of a phosphorylatable threonine (T688) (*Agulto et al., 2021*) in DCLK1.2 that is absent in the DCLK1.1 variant (*Figure 4B–C*), suggesting a possible exon duplication for adaptive regulation of DCLK1 function by phosphorylation. DCLK1.2 is the best-characterized isoform in terms of structure and function (*Cheng et al., 2022b*), and to compare it with DCLK1.1, we generated a series of C-terminal tail deletion mutants to evaluate how variations in the C-terminal tail contribute to isoform-specific DCLK1 functions.

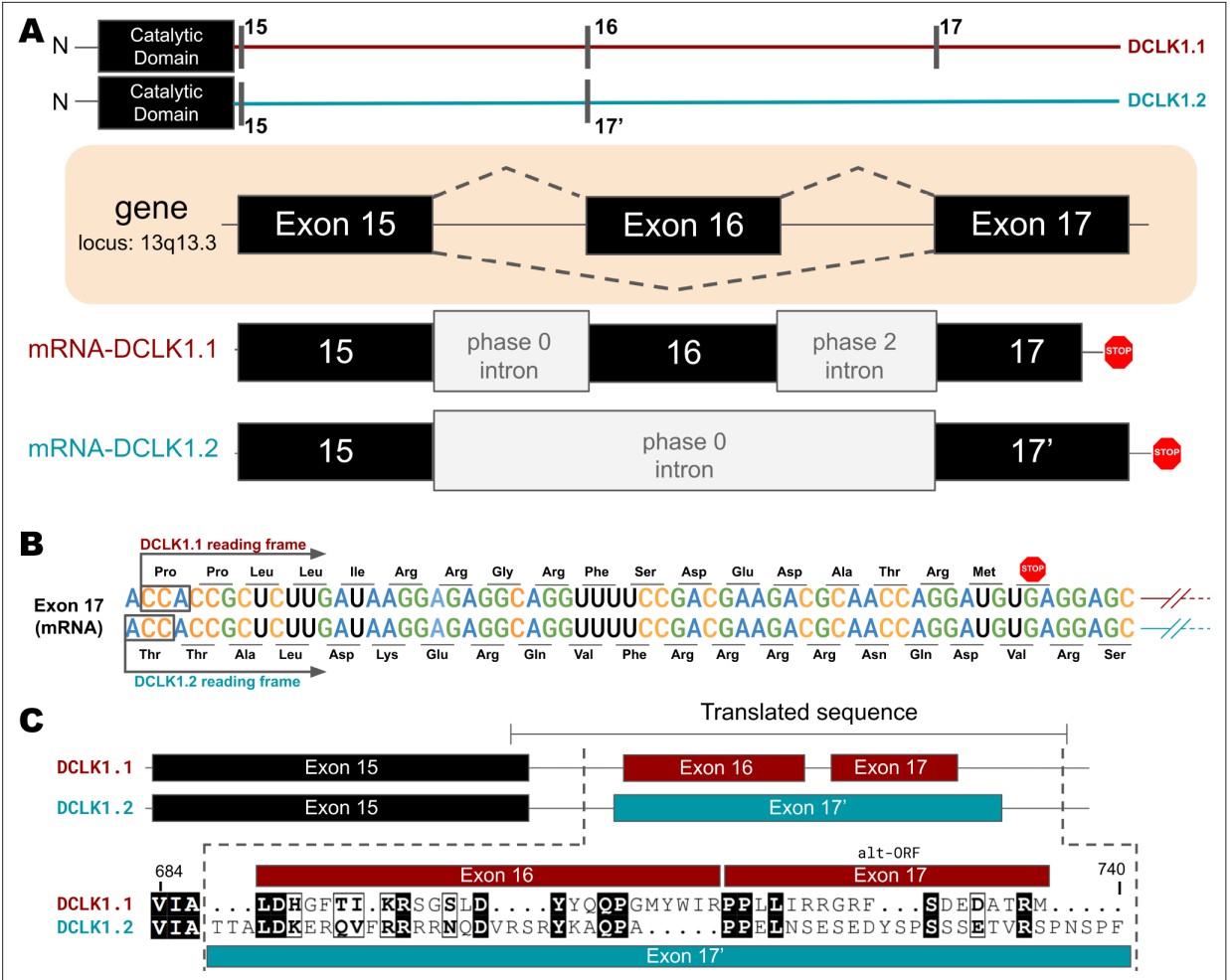

**Figure 4.** Transcriptional Variations in the C-terminal Tail of Human DCLK1 Isoforms: Gene Organization, Exon Alternation, and Protein Sequence Alignment. (**A**) Gene and intron-exon organization of DCLK1 human isoforms in the C-terminal tail. The DCLK1 gene is present on locus 13q13.3, and isoforms 1 and 3 contain an additional exon (exon 16) in the C-terminal tail that is absent in DCLK1.2. (**B**) A phase 2 intron results in the alternative transcript of exon 17 in isoform 1, translating a different open-reading frame and early stop codon, resulting in the shorter sequence. (**C**) Cartoon organization of the C-tail exons (exon 15, 16, and 17) of the DCLK1 isoforms, comparing the translated protein sequence alignment.

The online version of this article includes the following source data for figure 4:

**Source data 1.** Scipio output for intron-exon analysis of DCLK1.1.

**Source data 2.** Scipio output for intron-exon analysis of DCLK1.2.

**Source data 3.** Scipio output for intron-exon analysis of DCLK1.3.

**Source data 4.** Scipio output for intron-exon analysis of DCLK1.4.

## Isoform-specific variations encode changes in MD, thermostability, and catalytic activity in DCLK1

To study isoform-specific differences in the C-tail, we employed experimental techniques to compare protein stability and catalytic activity between purified DCLK1 proteins alongside MD simulations for DCLK1.1 and DCLK1.2 with different tail lengths. Isoforms 1.1 and 1.2 share identical sequences across the kinase domain and within the first 38 residues of the C-tail, and we used this information to design a new recombinant protein, termed $DCLK1_{cat}$ (residues 351–686). C-terminal to this totally conserved region, both isoforms possess extended tail segments, which includes the putative inhibitory-binding segment (IBS; residues 682–688) and an additional intrinsically disordered segment (IDS; residues 703-end). To study the role of the C-tail in modulating kinase stability and activity, we purified the $DCLK1_{cat}$, and C-tail containing (long and short) variants of each isoform, each of which lack the N-terminal DCX

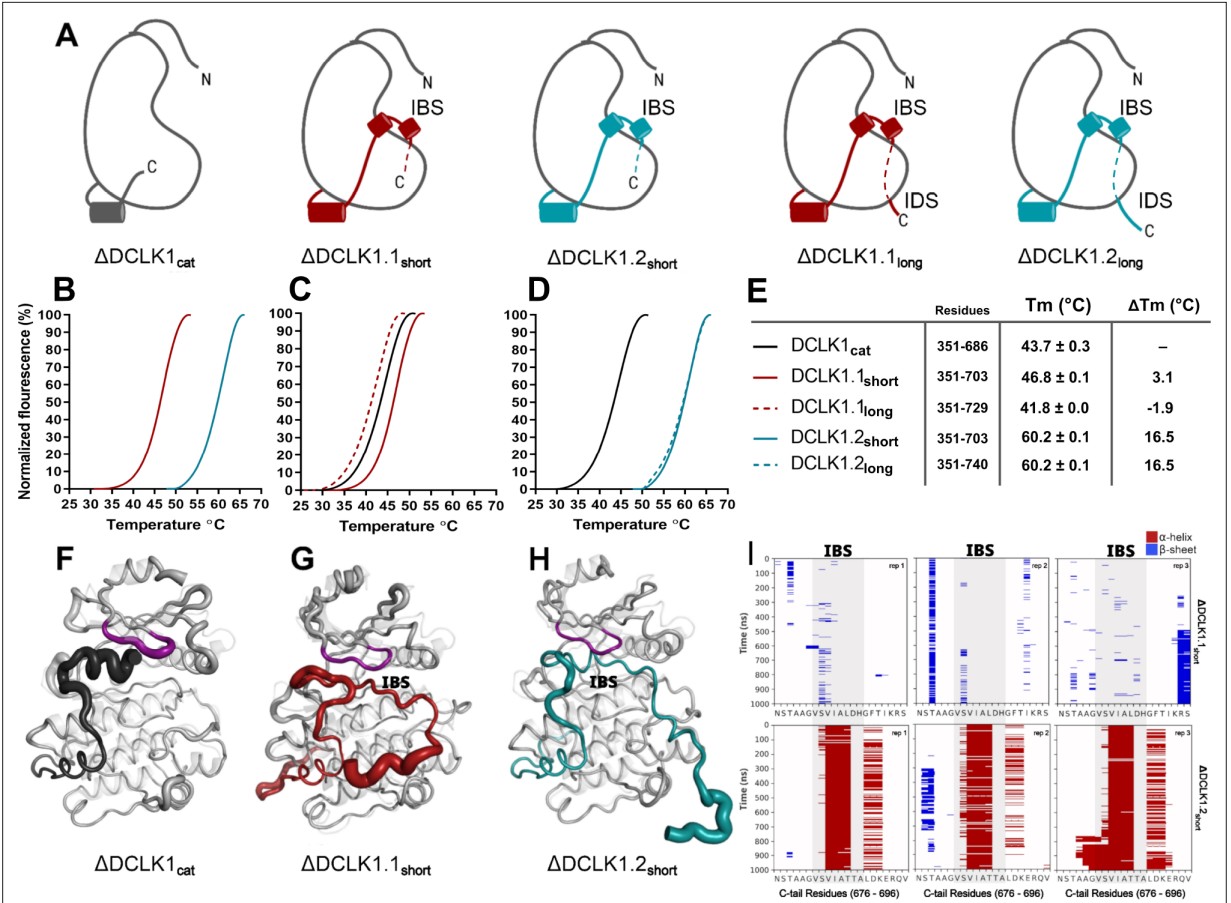

**Figure 5.** Structural Analysis, Thermal Stability, and Molecular Dynamics of DCLK1 Variants: Insights into the IBS and IDS Domains. (**A**) Cartoons of DCLK1 construct used in our assays, portraying the locations of the inhibitory-binding segment (IBS) and the intrinsically disordered segment (IDS). (**B–E**) Differential scanning fluorimetry (DSF) thermal denaturation profiling of the purified DCLK1 core catalytic domain, or tail-matched DCLK1.1 and DCLK1.2 proteins. Unfolding curves and changes in Tm values (ΔTm) for each protein relative to WT DCLK1cat are indicated. (**F–H**) B-factor structural representations of DCLK1short proteins shown in (**A**). The width of the region indicates the extent of flexibility based on averaged root mean square fluctuation (RMSF) data from three 1 µs molecular dynamics (MD) replicates. (**I**) DSSP analysis of three replicates of 1 µs MD simulations showing the residues surrounding the IBS in the C-tail of DCLK1.1short and DCLK1.2short. Blue indicates the presence of a beta-sheet or beta-bridge secondary structures and red indicates the presence of alpha-helical structures.

The online version of this article includes the following source data and figure supplement(s) for figure 5:

**Source data 1.** Root mean square fluctuation (RMSF) values for each mutant from all replicates of molecular dynamics (MD) simulations.

**Figure supplement 1.** SDS-PAGE and Coomassie blue staining of each DCLK1 protein.

**Figure supplement 1—source data 1.** Uncropped gel from SDS-PAGE of DCLK1 proteins.

**Figure supplement 1—source data 2.** PowerPoint file containing labels for Source Data 1.

**Figure supplement 2.** Comparative Interaction Analysis of K692 and H689 in DCLK1 C-tail Variants with DFG and HRD Aspartates.

**Figure supplement 3.** Root mean square fluctuation (RMSF) plots of molecular dynamics (MD) simulations of (**A**) ΔDCLK1.2 WT, (**B**) T688A, (**C**) pT688, and (**D**) T688E, where T688 is demarcated by a red line and the entire ΔDCLK1.2 C-tail is highlighted in light yellow.

domains (*Figure 5A*). SDS-PAGE demonstrated that protein preparations were essentially homogenous after affinity and gel filtration chromatography (*Figure 5—figure supplement 1*). The short forms of the recombinant proteins (DCLK1.1351-703 and DCLK1.2351-703) possess a partially truncated C-tail and were designed to match the amino acid sequence previously used to solve the structure of DCLK1.2 protein (*Cheng et al., 2022b*). Notably, these proteins exclude the IDS. The long forms of the DCLK1 proteins include the full-length C-tail for each isoform (DCLK1.1351-729 and DCLK1.2351-740) and incorporate IDS domains. We first performed comparative thermal shift analyses to quantify variance in thermal stability between the different purified proteins. When contrasting DCLK1.1short and

DCLK1.2$_{short}$ which do not differ in size or tail length but encode unique sets of amino acids in their partially truncated C-tail as a result of alternative splicing (*Figure 4*), we observed that DCLK1.2 was some 14°C more stable than DCLK1.1 (*Figure 5B*). When compared with DCLK1$_{cat}$, both DCLK1.1 short and long exhibited only subtle changes in thermal stability (*Figure 5C and E*), whereas both DCLK1.2 proteins (DCLK1.2$_{short}$ and DCLK1.2$_{long}$) were significantly stabilized (relative ΔTm >16°C, *Figure 5D and E*).

We next performed MD simulations to study the dynamics within the distinct DCLK1 C-tails that might explain the observed difference in protein stability. The crystal structure of DCLK1.2 (PDB: 6KYQ) was employed for the DCLK1.2$_{short}$ model and AlphaFold2 was used to model the other proteins (DCLK1$_{cat}$ and DCLK1.1$_{short}$). Comparison of the root mean square fluctuations (RMSF) of the two isoforms in three different replicates of MD simulations indicates strikingly different thermal fluctuations in the C-terminal tails and catalytic domains (*Figure 5—source data 1*). In particular, the IBS (between 682 and 688) is stably docked in the ATP-binding pocket in DCLK1.2 and an alpha-helical conformation is maintained during the microsecond time scale across different replicates (*Figure 5I*, bottom). In contrast, the IBS is more unstable in DCLK1.1, as indicated by high thermal fluctuations and a lack of secondary structure propensity (*Figure 5I*, top). A caveat to bear in mind is that DCLK1.1 is an AlphaFold2 model, which will also account for increased RMSF. Analysis of sequence variations and structural interactions provides additional insights into the differential dynamics of the two isoforms. The helical conformation of the IDS in DCK1.2 is maintained during the simulation due, in part, to a capping interaction with Thr 687, which is absent in DCLK1.1 due to the alternative splicing event detailed above. Likewise, another key residue in DCLK1.2, Lys 692, anchors the tail to the catalytic domain through directional salt bridges with the conserved aspartates (Asp 511 and Asp 533) in the HRD and DFG motifs (*Figure 5*, *Figure 5—figure supplement 2A*). These interactions are not observed in DCLK1.1 simulations because Lys 692 is substituted to a histidine (His 689), which is unable to form a corresponding interaction with the catalytic domain (*Figure 5*, *Figure 5—figure supplement 2B*). We also evaluated the effects of T688A (non-phosphorylated) or T688E (phosphomimetic) mutations through DCLK1 MD simulations and found that the two mutations slightly destabilize the tail relative to WT. Three replicates of the two mutants show increased RMSF of the tail region relative to WT DCLK1.1 (*Figure 5*, *Figure 5—figure supplement 3*). Either mutation was not sufficiently destabilizing on its own to unlatch the C-tail, and we hypothesize that other residues in addition to T688 are also likely to be important for contributing to conformational regulation of the kinase domain by the C-terminal tail. The variable docking of the C-tail within the kinase domains of the two DCLK1 isoforms, and the extent to which this contributes to more transient or stable autoinhibited states are explored in more detail in the next section.

## Residues contributing to the co-evolution and unique tethering of the C-terminal tail to the DCLK catalytic domain

To identify specific residues that contribute to the unique modes of DCLK regulation by the C-terminal tail, we performed statistical analysis of the evolutionary constraints acting on DCLK and related CAMK family sequences. We aligned the catalytic domain of DCLK and related CAMK sequences from diverse organisms and employed the Bayesian partitioning with pattern selection (BPPS) method (*Neuwald, 2014*) to identify residues that most distinguish DCLK sequences (foreground alignment in *Figure 6B*) from CAMK sequences (background alignment). Beyond the catalytic domain, DCLKs share sequence and structural similarities in the first helix of the tail (αR1 in CAMK1) (*Rosenberg et al., 2005*; *Goldberg et al., 1996*), with other CAMKs but share no detectable sequence similarity beyond this helical segment. DCLKs also share a CAMK-specific insert segment located between F and G helices in the catalytic domain, although the nature of residues conserved within the insert is unique to individual CAMK families (*Figure 6—figure supplement 1*). BPPS analysis revealed DCLK-specific constraints in different regions of the kinase domain, most notably, the ATP-binding G-loop, N terminus of the C-helix, the activation loop, and C-terminus of the F-helix (*Figure 6—figure supplement 2*).

Some of the most significant DCLK-specific residue constraints map to the ATP-binding G-loop (GDGNFA motif) (*Figure 6A–B*). In particular, Asp 398, Asn 400, and Ala 402 are unique to DCLKs as the corresponding residues are strikingly different in other CAMKs. Asp 398 is typically a charged residue (K/R) in other CAMKs while Asn 400 and Ala 402 are typically hydrophobic and polar residues,

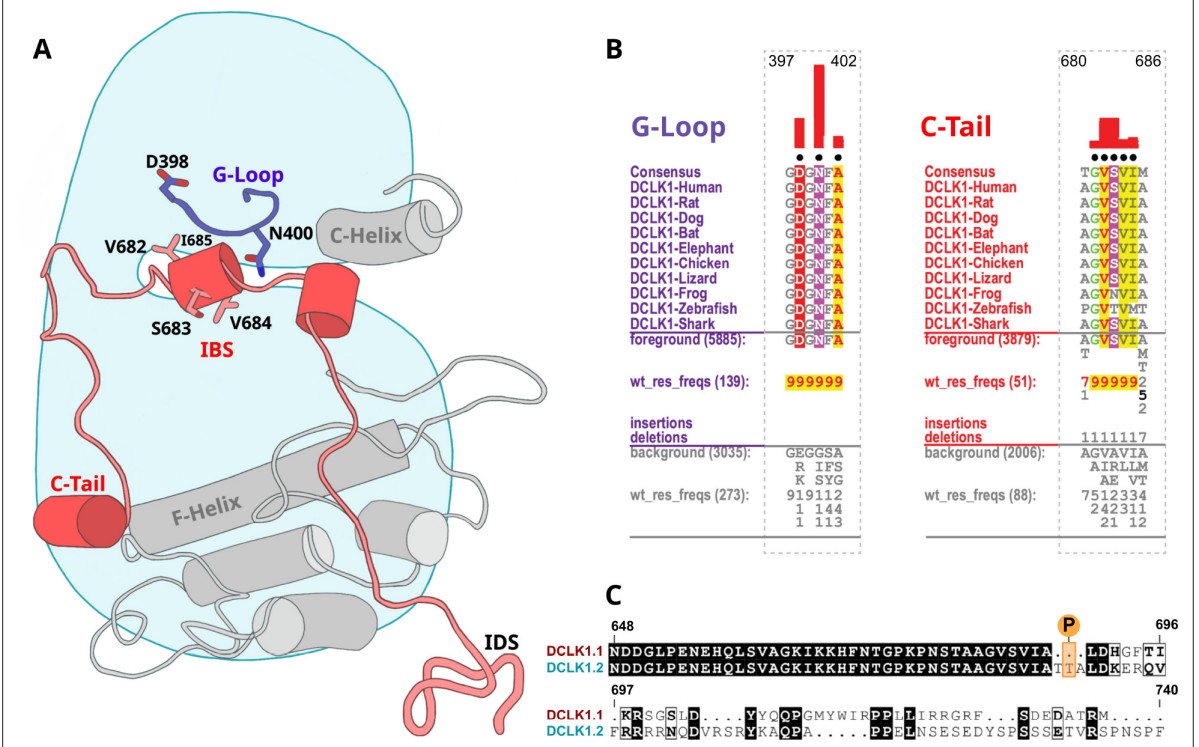

**Figure 6.** Identification of doublecortin-like kinase (DCLK)-specific constraints. (**A**) Cartoon of DCLK1.2 and the intrinsically disordered segment (IDS) with evolutionary constraints mapped to the kinase domain and C-tail. (**B**) Sequence constraints that distinguish DCLK1/2/3 sequences from closely related calcium calmodulin kinase (CAMK) sequences are shown in a contrast hierarchical alignment (CHA). The CHA shows DCLK1/2/3 sequences from diverse organisms as the display alignment. The foreground consists of DCLK sequences while the background alignment contains related CAMK sequences. The foreground and background alignments are shown as residue frequencies below the display alignment in integer tenths (*Manning et al., 2002*; *Agulto et al., 2021*; *Bayer and Schulman, 2019*; *Gógl et al., 2019*; *Berginski et al., 2021*; *Sossey-Alaoui and Srivastava, 1999*; *Ohmae et al., 2006*; *Couillard-Despres et al., 2005*; *Horesh et al., 1999*). The histogram (red) indicates the extent to which distinguishing residues in the foreground diverge from the corresponding position in the background alignment. Black dots indicate the alignment positions used by the Bayesian partitioning with pattern selection (BPPS) (*Neuwald, 2014*) procedure when classifying DCLK sequences from related CAMK sequences. Alignment number is based on the human DCLK1.2 sequence (UniProt ID: O15075-2). (**C**) Sequence alignment of human DCLK1 isoforms.

The online version of this article includes the following figure supplement(s) for figure 6:

**Figure supplement 1.** Calcium calmodulin kinase (CAMK)-specific insert (green) consistently making structural contacts (shown in surface representation) with the C-tail (red) across multiple CAMK families.

**Figure supplement 2.** Identification of doublecortin-like kinase (DCLK) family-specific constraints.

respectively (see residue frequencies in background alignment; *Figure 6B*). Notably, both Asn 400 and Asp 398 make direct interactions with residues in the C-tail either in the crystal structure or MD simulations (see below). Likewise, DCLK conserved residues in the C-helix and activation loop tether the C-terminal tail to functional regions of the kinase core, suggesting co-option of the DCLK catalytic domain to uniquely interact with the flanking cis-regulatory tail.

## An autoinhibitory ATP-mimic completes the C-spine and mimics the gamma phosphate of ATP

The most stable segment of the C-tail based on the B-factor and RMSF fluctuations in MD simulations is a unique region (682–688 in DCLK1.2) that docks into the ATP-binding pocket through both hydrophobic and hydrogen-bonding interactions. Remarkably, this peptide segment mimics the physiological ATP ligand, and stabilizes the catalytic domain through 'completion' of the hydrophobic catalytic spine (*Figure 7A*; *Zheng et al., 1993*). The residues that mimic adenosine and complete the C-spine of DCLK1 are Val 682, Val 684, and Ile 685 (PDB: 6KYQ), which are part of the alpha-helix that docks to the ATP-binding pocket (*Figure 7B*). Interestingly, based on our BPPS analyses, these C-tail residues are uniquely vertebrate DCLK1-specific pattern constraints. At the tail end of this

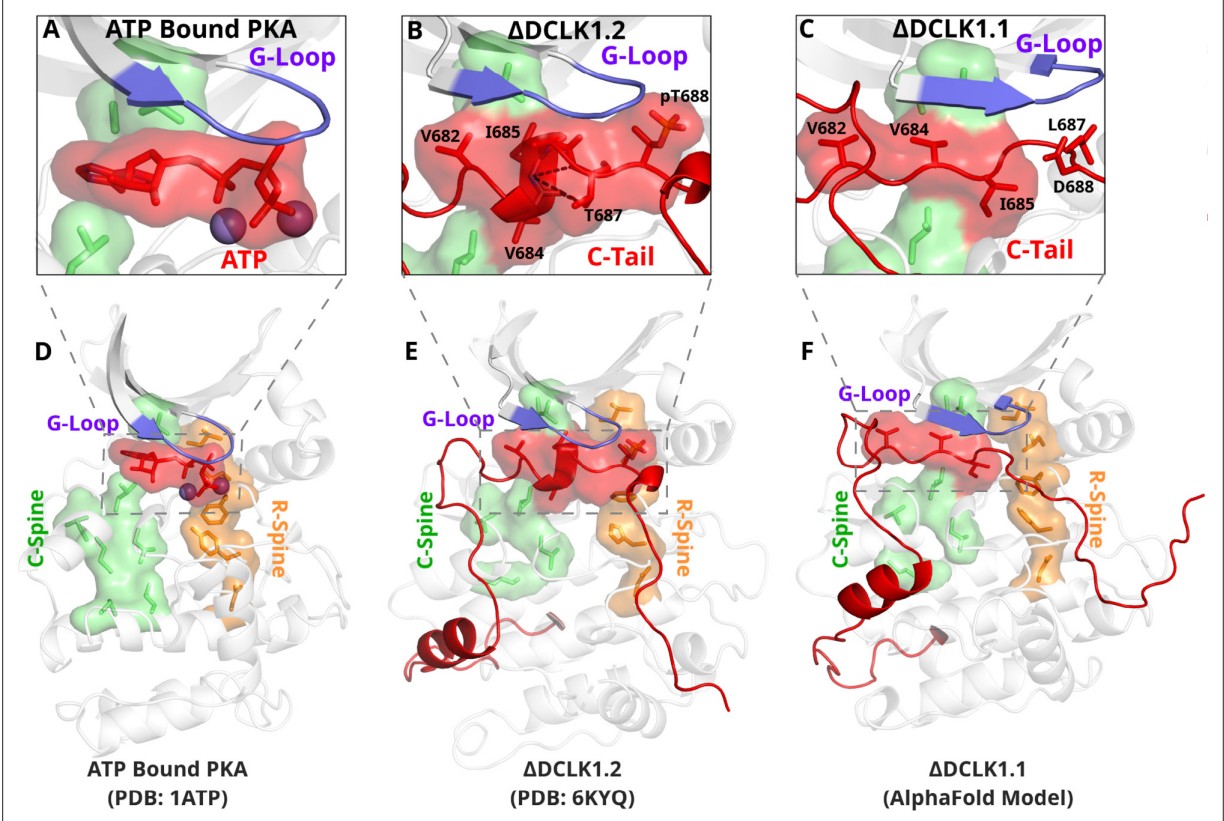

**Figure 7.** The DCLK1 C-tail 'completes' the regulatory C-spine (green). (**A**) PKA crystal structure (PDB: 1ATP) with bound ATP in red and Mg²⁺ in purple. The C-spine is completed by the adenine ring of ATP. The gamma phosphate of ATP hydrogen bonds with the second glycine of the G-loop. (**B**) DCLK1.2 crystal structure (PDB: 6KYQ) showing how the C-tail (red) docks underneath the pocket and mimics the ATP structure. The C-spine is completed by V682 and V684 in the C-tail and helical segments defined using DSSP are shown. T687 is also depicted making multiple hydrogen bonds with the backbone of V684 and I685 (dashed lines). (**C**) DCLK1.1 AlphaFold2 model showing an unstructured loop in the C-tail docking into the ATP-binding pocket, where V684 and I685 are predicted to complete the C-spine. The average per-residue confidence of the C-tail is 49%. (**D–F**) Zoomed-out versions of A–C, demonstrating how the DCLK1 C-tail docks into the ATP-binding cleft, akin to ATP in PKA.

The online version of this article includes the following figure supplement(s) for figure 7:

**Figure supplement 1.** Molecular dynamics of DCLK1 isoforms.

alpha-helix are two Thr residues, Thr 687 and Thr 688. As previously noted, these Thr residues mark the beginning of exon 17, and are one of the key variations between human DCLK1 isoforms, found only in DCLK1.2 variants.

Structural analysis and MD simulations reveal that Thr 687 in DCLK1.2 caps the stable alpha-helix that extends the C-spine (*Figure 7B*, *Figure 7—figure supplement 1B*). In comparison, the same region in DCLK1.1, which lacks Thr 687, is predicted to be unstructured. Upon phosphorylation, Thr 688 in DCLK1.2 can mimic the gamma phosphate of ATP by maintaining a stable hydrogen bonding distance with the backbone of the second glycine of the G-loop (G399) (*Figure 7B*, *Figure 7—figure supplement 1C*). We additionally observe that the sidechain of Asn 400, a DCLK-specific G-loop constraint, further stabilizes the phosphate group in pThr 688 through hydrogen bonding. As previously described, Thr 688 is unique to DCLK1.2. The lack of this functional site in DCK1.1 is correlated with increased RMSF and instability of the ATP-mimic segment in isoform 1 MDs (*Figure 5C–D*, *Figure 7—figure supplement 1B*). Comparatively, MD analysis of DCLK1.2 and a phosphothreonine-containing DCLK1.2 demonstrates reduced C-tail fluctuations, suggesting the potential regulatory involvement of Thr 688 phosphorylation for further modulation of the autoinhibited conformation (*Figure 7—figure supplement 1C*), consistent with previous findings (*Agulto et al., 2021*).

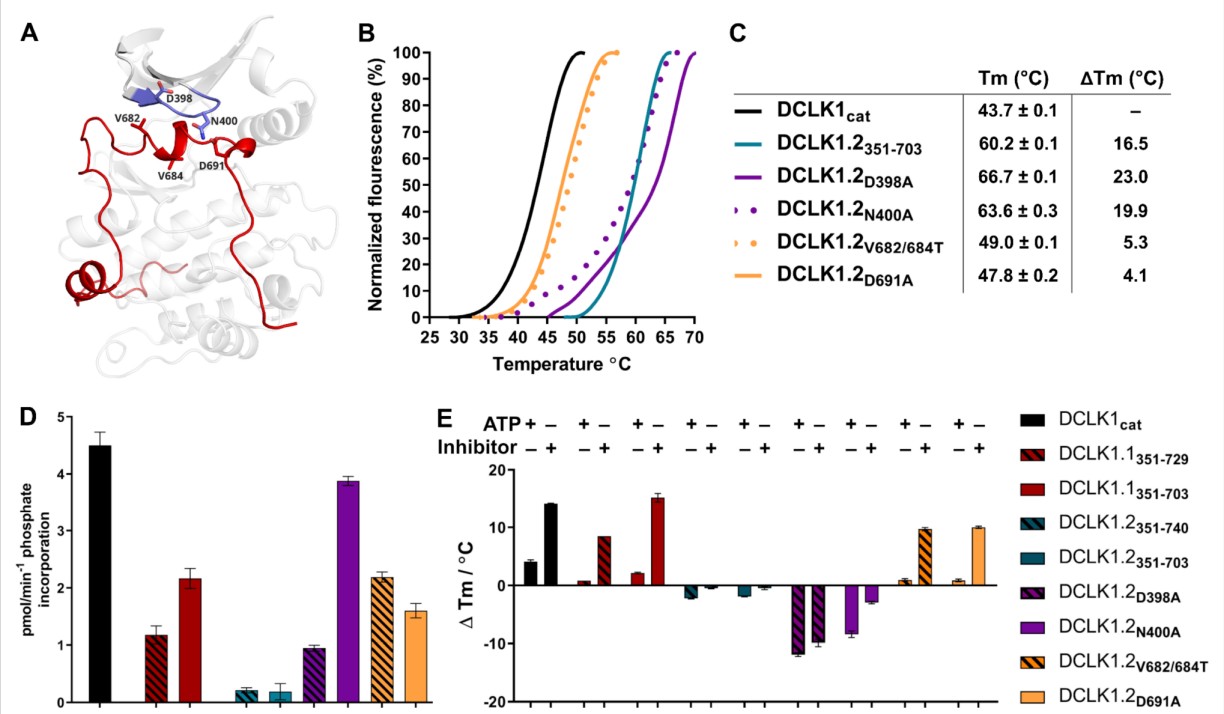

**Figure 8.** Structural and Functional Analysis of DCLK1.2 Variants: Insights into Amino Acid Modifications, Thermal Stability, and Kinase Activity.
(**A**) Structural depiction of DCLK1.2 (PDB: 6KYQ) showing the location of modified DCLK1 amino acids on the G-loop (purple) or C-tail (red). (**B–C**) Differential scanning fluorimetry assays depicting thermal denaturation profiles of each protein along with the calculated Tm value. (**D**) Kinase assays. DCLK1-dependent phosphate incorporation (pmol/min) into the DCLK1 peptide substrate was calculated for DCLK1$_{cat}$, long and short DCLK1.1, and the indicated DCLK1.2 variants. (**E**) Thermal stability analysis in the presence of ATP or DCLK1-IN-1 for DCLK1 proteins. For DCLK1.2, all proteins were generated in the DCLK1.2 short background.

The online version of this article includes the following figure supplement(s) for figure 8:

**Figure supplement 1.** Kinetic Analysis of DCLK1 Proteins: Phosphorylation Dynamics, ATP Affinity, and Derived Kinetic Parameters.

**Figure supplement 2.** Comprehensive Phosphorylation Site Mapping and Quantitative Analysis of DCLK Isoforms Using LC-MS/MS.

## Mutational analysis support isoform-specific allosteric control of catalytic activity by the C-terminal tail

To evaluate how sequence differences between DCLK1.1 and DCLK1.2 affected both thermal stability and catalytic potential, we generated targeted mutations at contact residues within the Gly-rich loop and C-tail of DCK1.1 and DCLK1.2 (at the indicted residues depicted in *Figure 8A*) which we predicted would disrupt or destabilize C-tail docking within the domain. All proteins were purified to near homogeneity by IMAC and SEC (*Figure 5—figure supplement 1*, *Figure 5—figure supplement 1—source data 1*), and the thermal stability of a panel of DCLK1.2 mutant and WT proteins were compared side-by-side with the DCLK1$_{cat}$ (*Figure 8B*). The ΔTm values obtained (*Figure 8C*) demonstrate that mutation of Asp 398 or Asn 400 in the Gly-rich loop are by themselves insufficient to destabilize DCLK1.2. In marked contrast, dual mutation of the hydrophobic pair of Val 682 and Val 684 residues to Thr, or mutation of the acidic tail residue Asp 691, resulted in a pronounced reduction in DCLK1.2 thermal stability. Moreover, the recorded Tm values for these latter two mutations quite closely resembled the Tm of DCLK1$_{cat}$ (which lacks the C-tail entirely), which is consistent with the uncoupling of the C-tail and a commensurate decrease in thermal stability associated with loss of this interaction. We next determined the catalytic activity of our recombinant DCLK1.1 and DCLK1.2 proteins side-by-side (*Figure 8D*, *Figure 8—figure supplement 1*). Although partially diminished in relation to DCLK1$_{cat}$, both DCLK1.1$_{short}$ (351–703) and DCLK1.1$_{long}$ (351–729) variants possess robust catalytic activity. This suggested ineffective ATP-competitive autoinhibition mediated by the C-tail segment of DCLK1.1 and is consistent with their closely matched Tm values to DCLK1$_{cat}$ (*Figure 5C*). Interestingly, both C-tail containing variants of DCLK1.1 (and particularly DCLK1.1$^{351-729}$) exhibited lower affinity for ATP

(inferred from $K_{M[ATP]}$ for peptide phosphorylation), which is consistent with partial occlusion of the ATP-binding pocket (*Figure 8—figure supplement 1*). In marked contrast, the detectable kinase activity for short (351–703) or long (351–740) DCLK1.2 proteins was significantly blunted compared to DCLK1$_{cat}$, exhibiting just ~5% of the activity of the catalytic domain alone, and consistently, the calculated $K_{M[ATP]}$ was ~4-fold higher compared to the catalytic domain lacking the C-tail. We also utilized autophosphorylation as a proxy for overall kinase activity. Quantitative tandem mass spectrometry (MS/MS) analysis of site-specific autophosphorylation within DCLK1.1$_{short}$ and DCLK1.2$_{short}$ demonstrate a marked reduction in the site-specific abundance of phosphate in DCLK1.2 when compared to DCLK1.1 at two separate sites that could be directly and accurately quantified by MS (S438 and S660, DCLK1.1 relative abundance set to 1, *Figure 8—figure supplement 2*). LC-MS/MS also indicated that several autophosphorylation sites identified in isoform 1 were absent in DCLK1.2 (Ser 683 and Thr 692, the latter of which is an amino acid that is unique to the C-tail of DCLK1.1, *Figure 8—figure supplement 2*). Interestingly, amino acid substitutions in the G-loop or the C-tail of DCLK1.2 designed to subvert C-tail and ATP site interactions also had major effects on DCLK1.2 phosphorylation and catalytic activity. DCLK1.2 D398A was activated some fivefold when compared to the WT form, whereas DCLK1.2 N400A was almost as active as the DCLK1$_{cat}$. Consistently, DCLK1.2 V682T/V684T and D691A proteins were also much more active than the WT form of DCLK1.2. Kinetic analysis also confirmed higher *Vmax* (but broadly similar $K_{M[ATP]}$) values for DCLK1.2 D398A and V682T/V684T relative to the WT protein (*Figure 8—figure supplement 1*). Moreover, comprehensive LC-MS/MS phosphosite mapping revealed a marked increase in the total number of phosphorylated amino acids in all of the mutant DCLK1.2 proteins, consistent with the enhanced catalytic activity of these proteins when compared to WT DCLK1.2 (*Figure 8—figure supplement 2*). Together, these observations confirm that targeted mutations are sufficient to relieve ATP-competitive C-tail autoinhibition by physical tail uncoupling; this model is also strongly supported by marked changes in the biophysical stability of the mutant proteins, particularly for V682/684T and D691A (*Figure 8B*).

To expand on this finding, we investigated the interaction of Mg:ATP or DCLK1-IN-1 to our panel of DCLK1.1 and -1.2 proteins (*Figure 8E*), using changes in thermal stability as a reporter of ligand binding. DCLK1$_{cat}$ (351–686), DCLK1.1$_{short}$ (351–703) and DCLK1.1$_{long}$ (351–729) proteins all behaved similarly in the presence of either Mg:ATP or DCLK1-IN-1, inducing marked stabilization. In contrast, DCLK1.2$_{short}$ (351–703) or DCLK1.2$_{long}$ (351–740) proteins registered negligible thermal shifts in the presence of the same concentration of either ligand, which is in line with the C-tail tightly occupying the ATP-binding site and obstructing their binding. Remarkably, D398A and N400A DCLK1.2, whose high basal Tm values (compared to DCLK1$_{cat}$) are consistent with stabilization by docking of the C-tail within the kinase domain, were markedly destabilized in the presence of either ligand. This suggests that incorporation of these G-loop mutations in isolation is insufficient to dislodge the C-tail, but rather that the stability of the interaction is compromised to the extent that either ATP or DCLK-IN-1 can competitively dislodge the bound tail from the ATP active site, resulting in a net destabilization caused by lack of tail engagement. This observation is corroborated by the results of our kinase assays, where both D398A and N400A mutants were more active than WT DCLK1.2, confirming appropriate ATP binding (a prerequisite for catalysis). Finally, DCLK1.2 V682T/V684T and D691A, which exhibit lower basal Tm values than the WT protein (indicating a loss of tail interaction), were both stabilized in a ligand-dependent manner to a similar degree to that observed for DCLK1$_{cat}$ and DCLK1.1 proteins (*Figure 8E*). Collectively, these observations clearly demonstrate that the C-tail section of DCLK1.2 can both stabilize the canonical DCLK kinase domain and inhibit kinase activity (by impeding the binding and structural coordination of Mg:ATP) much more effectively than that of DCLK1.1. Our targeted mutational analysis of key contact residues in DCLK1.2 also clearly shows that this is a consequence of specific amino acid interactions that are absent in DCLK1.1 due to alternative-splicing and subsequent sequence variation.

## Classification of DCLK regulatory segments

We synthesized our experimental findings by classifying the DCLK C-tail into six functional segments, based on interactions with different regions of the catalytic domain and their conservation between the C-tail splice variants in our analysis (*Figure 9*): First is an ATP-mimetic peptide segment (residues 682–688 in DCLK1.2) that readily mimics physiological ATP binding by completing the C-spine in the nucleotide-binding site. The inhibitory peptide also contains a phosphorylatable Thr residue, which

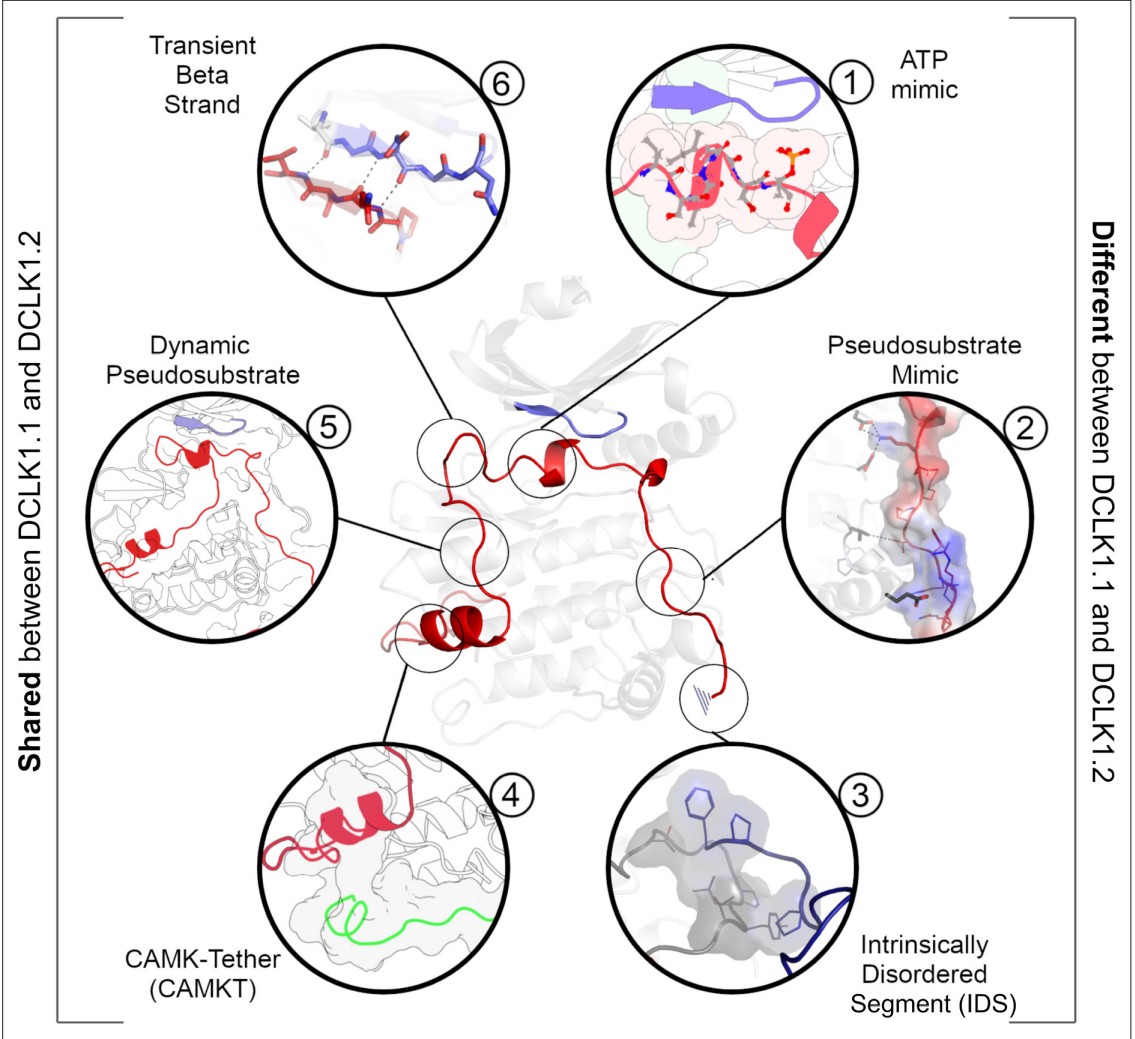

**Figure 9.** A DCLK1 C-tail can act as a multi-functional Swiss Army Knife, using six distinct segments for a variety of regulatory functions including mimicking ATP binding/association, stabilizing the G-loop, occluding the substrate-binding pocket, and packing against the kinase activation loop.

The online version of this article includes the following figure supplement(s) for figure 9:

**Figure supplement 1.** Intrinsic disorder prediction of DCLK1.2 C-tail using IUPRED3.

sits adjacent to the highly characteristic Gly-rich loop (GDGNFA, residues 396–402). Second, we define a pseudosubstrate mimic (residues 692–701), which interacts with the acidic HRD and DFG Asp side-chains and docks in the substrate pocket occluding substrate access. Third, at the C-terminus of the tail lies an IDS (residues 702–749, **Figure 9—figure supplement 1**), which packs dynamically against DCLK conserved residues in the kinase activation loop. Fourth, at the beginning of the C-tail lies a CAMK-tether (residues 654–664), a set of residues that pack against a CAMK-specific insert in the C-lobe. In many CAMK crystal structures, this insert makes multiple contacts with the F-helix and C-tail (**Figure 6—figure supplement 1**). Fifth, this is followed by a highly dynamic pseudosubstrate region (residues 672–678) that occludes the substrate pocket and will thus interfere with substrate phosphorylation. Sixth, a transient beta-strand is formed in DCLK1.2 through amino acid-specific sequences that help modulate and potentially strengthen binding of the C-tail in this isoform (**Figure 7—figure supplement 1A–B**). Collectively, these segments and their associated interaction sites demonstrate that co-evolution of the unique C-tail with the catalytic domain is the central hallmark of DCLK functional divergence, and that changes in these segments possess the ability to 'supercharge' catalytic output of the kinase. In particular, the variable C-terminal segments of the tail might contribute to isoform-specific functional specialization. The combinatorial diversity of events that modulate C-tail function may allow DCLKs to nimbly coordinate various tasks including ATP-binding, substrate-based

phosphorylation, regulation of DCX domain phosphorylation and structural disposition, kinase autoinhibition, and allosteric regulation. Isoform-specific variability provides additional nuance to regulatory and catalytic signaling events and may even contribute to differences in cellular localization (e.g. cytoplasm or nucleoplasm) and tissue-specific activity, enabling contextual DCLK regulation through these modular sequence segments.

## Discussion

The kinase domain is a conserved switch for phosphorylation-based catalytic regulation. Yet the complexity of cell signaling pathways demands other nuanced forms of regulation beyond binary 'on' or 'off' switch-based mechanisms. For many Ser/Thr kinases, including AGC and CAMK families, these distinct regulatory functions come from segments which flank the kinase domain, N- and C-terminal regions, which serve to modulate activity through allosteric activation, inhibition, or rheostatic behaviors that change based on environmental conditions (*Gógl et al., 2019*). In this study, we expand on our knowledge of allosteric diversity in the human kinome by revealing how alternative splicing of the DCLK1 C-tail contributes to isoform-specific behaviors, coupling regulation of catalytic output, phosphorylation, protein dynamics and stability, substrate binding, and protein-protein interactions. Our 'Swiss Army Knife' model for DCLK1 expands our view of allosteric regulation as not just a dynamic process facilitated by proteins, but one where adaptive genetic mechanisms, like differential splicing, dexterously tune isoform-specific functions for specific cellular signaling roles; in the case of DCLK1.1, this allows 'supercharging' of catalysis between splice variants due to key amino acid differences in the C-tail that are lacking in the DCLK1.2 isoform.

Multiple members of the human kinome have independently evolved C-tail regions that dock to the N- or C-lobe of the kinase domain in cis. In the case of the AGC kinases, the C-terminal tail is a very well-studied in cis modulatory element that serves to explain a variety of regulatory properties in this kinase sub-family (*Kannan et al., 2007*; *Romano et al., 2009*; *Baffi and Newton, 2022*; *Taylor and Kornev, 2011*). Classical deletion studies with members of the CAMK family have also revealed a cis-acting inhibitory element lying C-terminal to the catalytic domain of both CAMK1 (*Yokokura et al., 1995*) and CAMK2 (*Yang and Schulman, 1999*). More recent examples of C-tail functional diversity in CAMK family members are presented by the human pseudokinases TRIB1 and TRIB2, which employ C-tail sequences to either latch (and structurally restrict) the atypical kinase domain or to bind competitively to the ubiquitin E3 ligase COP1 (*Eyers, 2015*; *Eyers et al., 2017*). Functional disengagement of the TRIB1 or TRIB2 tail through deletion, mutagenesis, or small molecule binding has marked effects on pseudokinase conformation, intrinsic stability, and cellular transformation (*Foulkes et al., 2018*; *Harris et al., 2022*; *Keeshan et al., 2010*; *Murphy et al., 2015*).

### A novel pseudosubstrate region encoded by DCLK1

In addition to the marked differences between DCLK1 splice variants relevant to nucleotide binding, small molecule interactions, and catalysis, our work also reveals two unique pseudosubstrate segments present before and after the IBS. Before the IBS, we observe the formation of an anti-parallel transient beta-sheet with the beta1 strand in the catalytic domain (*Figure 5I*, *Figure 7—figure supplement 1A–B*). During the formation of this transient structure, the C-tail dynamically occludes part of the substrate-binding pocket. A beta-sheet is observed in all three MD replicates of DCLK1.1, but only in a single replicate of DCLK1.2 dynamic analysis. At the other end of the IBS is another pseudosubstrate segment whose structure and dynamics change as a result of alternative exon splicing. In DCLK1.2, the pseudosubstrate segment is stable, with an average RMSF of 1.8 Å, facilitated by key interactions from Lys 692, which coordinates acidic residues in the HRD and DFG motifs (*Figure 5—figure supplement 2A*). In DCLK1.1, Lys 692 is replaced by a His, which weakly coordinates with the HRD and DFG motifs, resulting in increased dynamics of the segment and an RMSF of 3.1 Å (*Figure 5*, *Figure 5—figure supplement 2B*). Together, residue variation between isoforms contributes to differences in stability and alteration of dynamics of the tail. HPCAL1 was recently proposed as a possible substrate that activates DCLK1 in a calcium-dependent manner, but it is unclear how it may bind DCLK1 (*Patel et al., 2021*). Because only exon 15 of the C-tail is conserved between the isoforms, it is possible that the location of binding occurs in this dynamic pseudosubstrate segment prior to the IBS, where

increased flexibility and occlusion of the substrate pocket is reflective of the absence of HPCAL1, or a similar CaM-like substrate.

## Discovery of the DCLK1 ATP-mimic region, a splice-variant specific regulatory module

Our structural analysis of DCLK1 reveals a remarkable structural mimic of ATP located in the C-tail, which differs markedly between DCLK1.1 and DLCK1.2 splice variants. We note for the first time that a set of three residues in the ATP-mimic, Val 682, Val 684, and Ile 685, are conserved across all isoforms of DCLK1 and DCLK2 (*Figure 1C*) and serve to extend the kinase C-spine. Mutation of these residues in DCLK1.2 uncouples tail binding and activates the kinase. Proximal to these residues are two Thr residues (Thr 687 and Thr 688), which are present in DCLK1.2, but absent in DCLK1.1. Based on published phosphoproteomics data, both Thr residues can be phosphorylated (*Agulto et al., 2021*) and are thus likely to be regulatory in DCLK1.2. Although we could not detect phosphorylation of either of these predicted regulatory sites in the WT form of DCLK1.2, we consistently observed pThr 688 in activated mutant DCLK1.2 variants (*Figure 8—figure supplement 2*). By analyzing DCLK1 dynamics using MD simulations, we observed multiple key interactions between the G-loop and the C-tail in DCLK1, such as dipole interactions with the second glycine in the G-loop by the phosphothreonine. In addition, Thr 687 contributes to increased stabilization of DCLK1.2 tail by forming a C-cap with the helical ATP-mimic segment. We aligned the intensively-studied protein kinase (PKA, PDB: 1ATP), a DCLK1.2 structure (PDB: 6KYQ), and frames of our MD trajectory, which demonstrate remarkable overlap of the ATP gamma phosphate and C-tail phosphothreonine, which seemingly acts as a mimic for the ATP co-factor. As phosphorylation is reported to lead to DCLK1 inhibition, this suggests a complex mechanism of regulation, in which the DCLK-specific constraints in the G-loop, the intrinsic flexibility of the C-tail, and threonine phosphorylation, by cis- or trans-mediated modification, systematically prevent hyperphosphorylation of the doublecortin domains and cellular effects. Somewhat paradoxically, we could only identify phosphorylated Thr 688 in activated DCLK1.2 mutants, but not in the autoinhibited (WT) version form. This suggests that the selected mutations exhibit a regulatory hierarchical dominance over inhibitory Thr 688 phosphorylation and are sufficient to liberate DCLK1.2 from its autoinhibited, C-tail-bound state. This also implies that phosphorylation of Thr 688 may only be minimally required for autoinhibition, especially given its association with the hyperactive variants obtained by mutagenesis (*Figure 8—figure supplements 1–2*).

Finally, we have evaluated the terminal residues in the DCLK1 C-tail, which are predicted to be intrinsically disordered. Side-by-side analysis of DCLK1.1 and DCLK1.2 in which this region is added from a common core terminating at residue 703 shows that increasing the length of the tail in both DCLK1.1 and DCLK1.2 has little additional effect on the inhibitory or stabilizing effects driven by the highly ordered tail regions that precede it. Kinase domains are regulated by IDRs in a multitude of ways, but the CAMK family is specifically enriched for adapted C-terminal extensions that, as we show here, can block the ATP and substrate binding, and enzymatically inactivate the kinase domain by occlusion of the activation loop through a flexible helical IDS on their C-tail (*Gógl et al., 2019*). We note that the DCLK1.2 kinase domain crystallizes in an 'active' closed conformation, despite binding of the C-tail in an autoinhibitory manner (*Cheng et al., 2022b*). Repeated packing motions of the IDS against the activation loop in all replicates of DCLK1.1 MD simulations suggest that tail may occlude the activation loop, similar to other CAMKs, possibly pointing to a mode of *cis* autoregulation. Conversely, AlphaFold2 predicts the placement of the DCX domains as adjacent to the IDS in both DCLK1.1 and DCLK1.2 (*Figure 1—figure supplement 1*). It is possible that, like other CAMKs, the IDS facilitates protein binding, whether to the DCX domains, calcium-modulated proteins, or other kinases.

## Evolutionary divergence and functional specialization of DCLKs

For the DCLK family as a whole, we discovered phylogenetic divergence between DCLK1 and -2 as a relatively recent event (*Figure 1A*) in which metazoan DCLK3 is the more ancestral DCLK gene from which DCLK1 and -2 emerged after duplication. Because of shared evolutionary constraints and the recent divergence between DCLK1 and -2, we surmise the functional specialization of the DCLK1 tail is shared between these paralogs. Moreover, we quantify key differences between human DCLK1.1 and -1.2 activity that are impacted by amino acid changes that contribute to the function of the C-tail.

The differences between DCLK1 isoforms 1 and 2 are generated by variations in exon splicing, which change both the C-tail protein sequence, and introduce or exclude potential phosphorylation sites. Expression of the highly autoinhibitable DCLK1.2 isoform is believed to be predominant in the brain during embryogenesis, although DCLK1.1 is also thought to be present in the adult brain (*Burgess and Reiner, 2002*). It is therefore possible that an altered ratio in DCLK1.1/1.2 expression, accompanied by changes in the requirement for DCLK1 autoregulation, is relevant for early neurogenesis. The overall sequence similarity at the protein level, despite the loss of a pair of putative phosphorylation sites, suggests a possible exon duplication in the C-tail, whereby polymorphisms have allowed for adaptive regulation during development and proliferation. Indeed, we also speculate that the induced expression of DCLK1.2 splice variants in multiple cancer subtypes (*Qu et al., 2019*) is likely to be indicative of a survival and drug resistance role that could be targetable with new types of small molecule. Although nanomolar DCLK1 ATP site inhibitors such as DCLK1-IN-1 have been developed that can bind tightly to the DCLK1 ATP site (*Figure 3* and *Figure 3—figure supplement 1*), the 'problematic' existence of human DCLK1.1 and DCLK1.2 splice variants with distinct autoinhibitory properties may present a challenge to compound engagement in the cell, where relief of autoinhibition through C-tail undocking in DCLK1.2 is likely to require a high concentration of compound in order to compete and disengage interactions at the ATP site. Indeed, although potent chemical DCLK1 inhibitors such as DCLK1-IN-1 are known to influence DCLK1 autophosphorylation and cell motility, they have relatively modest effects in cells in terms of cytotoxicity (*Ferguson et al., 2020*; *Ding et al., 2021*). Therefore, we propose that the dual inhibitory effects of the C-tail and the transmission of this information to adjacent DCX domains, which control adaptive cellular phenotypes such as EMT in cancer cells (*Major et al., 1985*), may make allosteric classes of DCLK1 inhibitor a preferred therapeutic option, especially if they can be tailored specifically toward DCLK1.1 or DCLK1.2, whose autoregulation is different in terms of the varied molecular details we have uncovered here.

## Materials and methods
### Ortholog identification
To identify orthologs, we used the software KinOrtho (*Huang et al., 2021*) to query one-to-one orthologous relationships for DCLK1/2/3 across the proteome. After collation of the various orthologs, we parsed the sequence data for taxonomic information and classified each sequence by family. We further separated human DCLK1 into each unique isoform and aligned them.

### Phylogenetic analysis
We identified diverse DCLK orthologs from the UniProt database (*UniProt Consortium, 2021*) using an profile-based approach (*Neuwald, 2009*). From this dataset, we manually curated a taxonomically diverse set of DCLK orthologs composed of 36 sequences spanning 16 model organisms. These sequences were used to generate a maximum-likelihood phylogenetic tree using IQTREE version 1.6.12 (*Nguyen et al., 2015b*). Branch support values were generated using ultrafast bootstrap with 1000 resamples (*Hoang et al., 2018*). The consensus tree was selected as the final tree. The optimal substitution model for our final topology was determined to be LG (*Le and Gascuel, 2008*) with invariable sites and discrete gamma model (*Gu et al., 1995*) based on the Bayesian information criterion as determined by ModelFinder (*Kalyaanamoorthy et al., 2017*). We rooted our final tree against an outgroup of 17 closely related human CAMK kinases using ETE Toolkit version 3.1.2 (*Huerta-Cepas et al., 2016*).

### Sequence and structure analysis
MAFFT (*Katoh et al., 2002*) generated multiple sequence alignments were fed into the BPPS tool to determine evolutionarily conserved and functionally significant residues (*Neuwald, 2014*). Constraints mapped onto AlphaFold-predicted structures were visualized in PyMOL to analyze biochemical interactions.

### Rosetta loop modeling
Loop modeling was performed on the crystal structure (6KYQ) using the kinematic loop modeling protocol (*Mandell et al., 2009*) to model missing residues. Following this, the structure underwent

five cycles of rotamer repacking and minimization using the Rosetta Fast-relax protocol (*Tyka et al., 2011*).

## DSSP analysis

To analyze changes in secondary structure over our MDs, we employed the DSSP (*Kabsch and Sander, 1983*) command in GROMACS. This produces an output that contains an array of secondary structure values against each residue. The MDAnalysis python module (*Michaud-Agrawal et al., 2011*) was used to plot these values.

## Molecular dynamics

PDB constructs were generated by retrieving structural models RCSB and the $\mathbb{A}$lphaFold2 database. Post-translational modifications were performed in PyMOL using the PyTMs plugin. All structures were solvated using the TIP3P water model (*Jorgensen et al., 1983*). Energy minimization was run for a maximum of 10,000 steps, performed using the steepest descent algorithm, followed by the conjugate gradient algorithm. The system was heated from 0 K to a temperature of 300 K. After two equilibration steps that each lasted 20 ps, 1-µs-long simulations were run at a 2 fs timestep. Long-range electrostatics were calculated via particle mesh Ewald algorithms using the GROMACS MD engine (*Pronk et al., 2013*). We utilized the CHARMM36 force field (*Huang and MacKerell, 2013*). The resulting output was visualized using VMD 1.9.3 (*Humphrey et al., 1996*). All MD analysis was conducted using scripts coded in Python using the MDAnalysis module (*Michaud-Agrawal et al., 2011*).

## Computational mutational analysis

Cartesian ddG in Rosetta (*Park et al., 2016*) was utilized to predict potential stabilizing and destabilizing mutations in the enzyme structure. We performed three replicates per mutation and averaged the Rosetta energies. All mutant energies were then subtracted by the WT Rosetta energy to generate a panel of ddG values relative to WT. Combined with our sequence analyses, we mutated kinase and DCLK-specific constraints to identify destabilizing interactions in the C-tail.

## Exon-intron boundary mapping

The precise gene structure of DCLK1 isoforms were mapped onto the human genome with each isoform used as a query protein sequence in order to generate exon-intron borders. This was achieved using Scipio (version 1.4.1) (*Keller et al., 2008*) with default settings. Exons were numbered based on Ensembl annotations (*Cunningham et al., 2022*). The translation of each annotated gene sequence to protein sequence was provided with the output file (*Figure 4—source data 1–4*).

## DCLK1 cloning and recombinant protein expression

6His-DCLK1 catalytic domain (351–686), DCLK1.1 351–703 (short C-tail) or 351–729 (full C-tail), DCLK1.2 351–703 (short C-tail) or 351–740 (full C-tail), and DCLK1.2 351–703 containing D398A, N400A, V682T/V684T, or D691A substitutions were synthesized by Twist Biosciences in pET28a. 6His-GST-(3C) DCLK1.1 351–689 was amplified by PCR and cloned into pOPINJ. Kinase dead, D533A 6His-GST-(3C) DCLK1.1 351–689 was generated by PCR-based site-directed mutagenesis (*Figure 3—source data 3*). All plasmids were sequenced prior to their use in protein expression studies. All proteins, including 6His-GST-(3C) DCLK1 351–689, with a 3C protease cleavable affinity tag, were expressed in BL21(DE3)pLysS *Escherichia coli* (Novagen) and purified by affinity and SEC. The short N-terminal 6His affinity tag present on all other DCLK1 proteins described in this paper was left in situ on recombinant proteins, since it does not appear to interfere with differential scanning fluorimetry (DSF), biochemical interactions, or catalysis. For analytical SEC (*Figure 3—figure supplement 1—source data 1*, *Figure 3—figure supplement 2—source data 1*), 1 mg of each DCLK1 protein was assayed on a Superdex 200 Increase 10/300 GL (Cytiva), and the eluted fractions were also analyzed by SDS-PAGE and Coomassie blue staining to confirm composition. The molecular weight standards were loaded in a mixture of 200 µg of bovine serum albumin, carbonic anhydrase, and alcohol dehydrogenase each.

## Mass spectrometry

Purified DCLK1 proteins (5 µg) were diluted (~40-fold) in 100 mM ammonium bicarbonate pH 8.0 and reduced (DTT) and alkylated iodoacetamide, as previously described (*Ferries et al., 2017*), and digested with a 25:1 (wt/wt) trypsin gold (Promega) at 37°C for 18 hr with gentle agitation. Digests were then subjected to strong cation exchange chromatography using in-house packed stage tip clean-up (*Daly et al., 2021*). Dried tryptic peptides were solubilized in 20 µl of 3% (vol/vol) acetonitrile and 0.1% (vol/vol) TFA in water, sonicated for 10 min, and centrifuged at 13,000 × *g* for 10 min at 4°C and supernatant collected. LC-MS/MS separation was performed using an Ultimate 3000 nano system (Dionex), over a 60 min gradient (*Ferries et al., 2017*). Briefly, samples were loaded at a rate of 12 µl/min onto a trapping column (PepMap100, C18, 300 µm×5 mm) in loading buffer (3% [vol/vol] acetonitrile, 0.1% [vol/vol] TFA). Samples were then resolved on an analytical column (Easy-Spray C18 75 µm×500 mm, 2 µm bead diameter column) using a gradient of 97% A (0.1% [vol/vol] formic acid): 3% B (80% [vol/vol] acetonitrile, 0.1% [vol/vol] formic acid) to 80% B over 30 min at a flow rate of 300 nl/min. All data acquisition was performed using a Fusion Lumos Tribrid mass spectrometer (Thermo Scientific). Samples were injected twice with either higher-energy C-trap dissociation (HCD) fragmentation (set at 32% normalized collision energy [NCE]) or electron transfer dissociation with supplemental 30% NCE HCD (EThcD) for 2+ to 4+ charge states using a top 3 s top speed mode. MS1 spectra were acquired at a 120 K resolution (at 200 $m/z$), over a range of 300–2000 $m/z$, normalized AGC target = 50%, maximum injection time = 50 ms. MS2 spectra were acquired at a 30 K resolution (at 200 $m/z$), AGC target = standard, maximum injection time = dynamic. A dynamic exclusion window of 20 s was applied at a 10 ppm mass tolerance. Data was analyzed by Proteome Discoverer 2.4 in conjunction with the MASCOT search engine using a custom database of the UniProt *E. coli* reviewed database (updated January 2023) with the DCLK1 mutant variant amino acid sequences manually added, and using the search parameters: fixed modification = carbamidomethylation (C), variable modifications = oxidation (M) and phospho (S/T/Y), MS1 mass tolerance = 10 ppm, MS2 mass tolerance = 0.01 Da, and the *ptm*RS node on; set to a score >99.0. For HCD data, instrument type = electrospray ionization-Fourier transform ion cyclotron resonance, for EThcD data, instrument type = EThcD. For label-free relative quantification of phosphopeptide abundances of the different DCLK1 variants, the minora feature detector was active and set to calculate the area under the curve for peptide $m/z$ ions. Abundance of phosphopeptide ions were normalized against the total protein abundance (determined by the HI3 method [*Silva et al., 2006*], as in the minora feature detector node) to account for potential protein load variability during analysis.

## DCLK1 DSF

Thermal shift assays were performed using DSF in a StepOnePlus Real-Time PCR machine (Life Technologies) in combination with Sypro-Orange dye (Invitrogen) and a thermal ramping protocol (0.3°C per minute between 25°C and 94°C). Recombinant DCLK1 proteins were assayed at a final concentration of 5 µM in 50 mM Tris-HCl (pH 7.4) and 100 mM NaCl in the presence or absence of the indicated concentrations of ligand (ATP or Mg:ATP) or DCLK1 inhibitor compounds, with final DMSO concentrations never higher than 4% (vol/vol). Thermal melting data were processed using the Boltzmann equation to generate sigmoidal denaturation curves, and average $T_m/\Delta T_m$ values were calculated as described using GraphPad Prism software, as previously described, from three technical repeats (*Byrne et al., 2020a*).

## DCLK1 kinase assays

DCLK1 peptide-based enzyme assays (*Byrne et al., 2016*; *Omar et al., 2023*) were carried out using the LabChip EZ Reader platform, which monitors and quantifies real-time phosphorylation-induced changes in the mobility of the fluorescently labeled DCLK1 peptide substrate 5-FAM-KKALRRQETVDAL-CONH$_2$. To assess DCLK1 catalytic domains, or DCLK1.1 or DCLK1.2 variants, 100 ng of purified protein were incubated with a high (1 mM) concentration of ATP (to mimic cellular levels of nucleotide) and 2 µM of the fluorescent substrate in 25 mM HEPES (pH 7.4), 5 mM MgCl$_2$, and 0.001% (vol/vol) Brij 35. DCLK1-IN-1 and DCLK1-NEG (kind gifts from Dr Fleur Ferguson, UCSF) enzyme inhibition was quantified under identical assay conditions in the presence of 10 µM of each compound. Assays are either reported as rates (pmoles/min phosphate incorporation) during linear phosphate incorporation (e.g. total substrate phosphorylation limited to <20–30% to prevent

ATP depletion and to ensure assay linearity) or presented as time-dependent percentage substrate phosphorylation (kinetic mode). Rates of substrate phosphorylation (pmol phosphate incorporation per minute) were determined using a fixed amount of kinase and linear regression analysis with GraphPad Prism software; $V_{max}$ and $K_{M[ATP]}$ values were calculated at 2 µM substrate peptide concentration, as previously described (*McSkimming et al., 2016*). Rates are normalized to enzyme concentration and all enzyme rate and kinetic data are presented as mean and SD of four technical replicates.

## Acknowledgements

Funding from NK (grant no. R35 GM139656) is acknowledged. PAE acknowledges funding from a University of Liverpool BBSRC IAA award. AV acknowledges funding from ARCS Foundation. DPB, LAD, PAE, and CEE also acknowledge BBSRC grants BB/S018514/1, BB/N021703/1, and BB/X002780/1 and North West Cancer Research (NWCR) grant CR1208. EEF thanks the MRC for a DiMeN DTP studentship (No. 1961582). The content is solely the responsibility of the authors and does not necessarily represent the official views of the National Institutes of Health.

## Additional information

### Funding

| Funder | Grant reference number | Author |
|---|---|---|
| National Institute of General Medical Sciences | GM139656 | Natarajan Kannan |
| North West Cancer Research | CR1208 | Leonard A Daly<br>Claire E Eyers<br>Patrick A Eyers<br>Dominic P Byrne |
| Biotechnology and Biological Sciences Research Council | BB/S018514/1 | Dominic P Byrne<br>Leonard A Daly<br>Patrick A Eyers<br>Claire E Eyers |
| Biotechnology and Biological Sciences Research Council | BB/N021703/1 | Dominic P Byrne<br>Leonard A Daly<br>Patrick A Eyers<br>Claire E Eyers |
| Biotechnology and Biological Sciences Research Council | BB/X002780/1 | Dominic P Byrne<br>Leonard A Daly<br>Patrick A Eyers<br>Claire E Eyers |

The funders had no role in study design, data collection and interpretation, or the decision to submit the work for publication.

### Author contributions

Aarya Venkat, Conceptualization, Data curation, Software, Formal analysis, Funding acquisition, Validation, Investigation, Visualization, Methodology, Writing – original draft, Writing – review and editing; Grace Watterson, Data curation, Formal analysis, Validation, Investigation, Visualization, Methodology, Writing – original draft, Writing – review and editing; Dominic P Byrne, Formal analysis, Funding acquisition, Validation, Investigation, Methodology, Writing – original draft, Writing – review and editing; Brady O'Boyle, Data curation, Formal analysis, Investigation, Visualization, Methodology, Writing – review and editing; Safal Shrestha, Formal analysis, Investigation, Visualization, Writing – review and editing; Nathan Gravel, Validation, Investigation, Writing – review and editing; Emma E Fairweather, Leonard A Daly, Funding acquisition, Investigation, Writing – review and editing; Claire Bunn, Writing – review and editing; Wayland Yeung, Data curation, Writing – review and editing; Ishan Aggarwal, Samiksha Katiyar, Investigation, Writing – review and editing; Claire E Eyers, Funding acquisition, Methodology, Writing – review and editing; Patrick A Eyers, Resources, Data curation, Supervision, Funding acquisition, Methodology, Writing – original draft, Writing – review and editing;

Natarajan Kannan, Conceptualization, Resources, Supervision, Funding acquisition, Writing – original draft, Project administration, Writing – review and editing

**Author ORCIDs**
Aarya Venkat http://orcid.org/0000-0002-8793-4097
Claire E Eyers https://orcid.org/0000-0002-3223-5926
Patrick A Eyers http://orcid.org/0000-0002-9220-2966
Natarajan Kannan http://orcid.org/0000-0002-2833-8375

Reviewer #1 (Public Review): https://doi.org/10.7554/eLife.87958.3.sa1
Reviewer #2 (Public Review): https://doi.org/10.7554/eLife.87958.3.sa2
Author Response https://doi.org/10.7554/eLife.87958.3.sa3

## Additional files

### Supplementary files
• MDAR checklist

### Data availability
All data generated in this study are included within the manuscript. Source data are provided for each figure. MD simulations and associated data have been deposited to Dryad (https://doi.org/10.5061/dryad.8931zcrxb).

The following dataset was generated:

| Author(s) | Year | Dataset title | Dataset URL | Database and Identifier |
|---|---|---|---|---|
| Venkat A, Watterson G, Kannan N | 2023 | Molecular Dynamics Simulations and associated data for Mechanistic and evolutionary insights into isoform-specific 'supercharging' in DCLK family kinases | https://doi.org/10.5061/dryad.8931zcrxb | Dryad Digital Repository, 10.5061/dryad.8931zcrxb |

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
