## [Editor Report · eLife assessment]

This **important** study expands on current knowledge of allosteric diversity in the human kinome by C-terminal splicing variants using as a paradigm DCLK1. The authors provide **convincing** evolutionary and some mechanistic evidence how C-terminal isoform specific variants generated by alternative splicing can regulate catalytic activity by means of coupling specific phosphorylation sites to dynamical and conformational changes controlling active site and substrate pocket occupancy, as well as protein-protein interactions. The data will be of interest to researchers in the kinase and signal transduction field.

---

## [Referee Report · Reviewer #1 (Public Review)]

In the study by Venkat et al. the authors expand the current knowledge of allosteric diversity in the human kinome by c-terminal splicing variants using as a paradigm DCLK1. In this work, the authors provide evolutionary and some mechanistic evidence about how c-terminal isoform specific variants generated by alternative splicing can regulate catalytic activity by means of coupling specific phosphorylation sites to dynamical and conformational changes controlling active site and substrate pocket occupancy, as well as interfering with protein-protein interacting interfaces that altogether provides evidence of c-terminal isoform specific regulation of the catalytic activity in protein kinases.

The paper is overall well written, the rationale and the fundamental questions are clear and well explained, the evolutionary and MD analyses are very detailed and well explained. Overall I think this is a study that brings some new aspects and concepts that are important for the protein kinase field, in particular the allosteric regulation of the catalytic core by c-terminal segments, and how evolutionary cues generate more sophisticated mechanisms of allosteric control in protein kinases.

Current submission: I have read and gone through the revised manuscript and the rebuttal letter and I confirm that the authors did an excellent job answering all the comments satisfactorily.

---

## [Referee Report · Reviewer #2 (Public Review)]

In this study, the authors explore the structure/function of the DCLK kinases, most specifically DCLK1 as it is the most studied to date. Recently, the C-terminal domain has garnered attention as it was found to regulate the kinase domain, however, the different isoforms retain additional amino acid sequences with as-yet-undefined functions. The authors provide an evolutionary and biochemical characterization of these regions and provide evidence for some functionality for these additional C-terminal sequences. While these experiments are informative they do require that the protein is soluble and not membrane-bound as has been suggested to be important for functionality in other studies. Still, this is a major contribution to understanding the structure/function of these proteins that will be important in future experimental designs.

---

## [Author Response]

The following is the authors’ response to the original reviews.

**eLife assessment**
This important study expands on current knowledge of allosteric diversity in the human kinome by C-terminal splicing variants using as a paradigm DCLK1. The authors provide solid evolutionary and some mechanistic evidence how C-terminal isoform specific variants generated by alternative splicing can regulate catalytic activity by means of coupling specific phosphorylation sites to dynamical and conformational changes controlling active site and substrate pocket occupancy, as well as protein-protein interactions. The data will be of interest to researchers in the kinase and signal transduction field.

We thank the editor for coordinating the review of our manuscript and the reviewers for their valuable feedback. We have significantly revised the manuscript in response to the reviewer’s comments. Our point-by-point response to each comment is present below. We have uploaded both a clean draft of our revised manuscript as well as a version with the revisions highlighted in yellow. We hope the revised manuscript is now acceptable for publication in eLife. We have additionally updated the preprint on bioRxiv and have included the link: We thank the editor for coordinating the review of our manuscript and the reviewers for their valuable feedback. We have significantly revised the manuscript in response to the reviewer’s comments. Our point-by-point response to each comment is present below. We have uploaded both a clean draft of our revised manuscript as well as a version with the revisions highlighted in yellow. We hope the revised manuscript is now acceptable for publication in eLife. We have additionally updated the preprint on biorxiv and have included the link here: https://www.biorxiv.org/content/10.1101/2023.03.29.534689v2.

**Reviewer #1**
SummaryIn the study by Venkat et al. the authors expand the current knowledge of allosteric diversity in the human kinome by c-terminal splicing variants using as a paradigm DCLK1. In this work, the authors provide evolutionary and some mechanistic evidence about how c-terminal isoform specific variants generated by alternative splicing can regulate catalytic activity by means of coupling specific phosphorylation sites to dynamical and conformational changes controlling active site and substrate pocket occupancy, as well as interfering with protein-protein interacting interfaces that altogether provides evidence of c-terminal isoform specific regulation of the catalytic activity in protein kinases.The paper is overall well written, the rationale and the fundamental questions are clear and well explained, the evolutionary and MD analyses are very detailed and well explained. The methodology applied in terms of the biochemical and biophysical tools falls a bit short in some places and some comments and suggestions are given in this respect. If the authors could monitor somehow protein auto-phosphorylation as a functional readout would be very useful by means of using phospho-specific antibodies to monitor activity. Overall I think this is a study that brings some new aspects and concepts that are important for the protein kinase field, in particular the allosteric regulation of the catalytic core by c-terminal segments, and how evolutionary cues generate more sophisticated mechanisms of allosteric control in protein kinases. However a revision would be recommended.Major CommentsThe authors explain in the introduction the role of T688 autophosphorylation site in the function of DCLK1.2. This site when phosphorylated have a detrimental impact on catalytic activity and inhibits phosphorylation of the DCX domain. allowing the interaction with microtubules. In the paper they show how this site is generated by alternative splicing and intron skipping in DCLK1.2. However there is no further functional evidence along the functional experiments presented in this study.1. What is the effect of a non-phosphorylable T688 mutant in terms of stability and enzymatic activity? What would be the impact of this mutant in the overall auto-phosphorylation reaction?

The role of T688 phosphorylation on DCLK1 functions has been explored in previous studies (Agulto et al, 2020: PMID: 34310279), although only relevant to DCLK1.2 splice variants, since this site is lacking in DCLK1.1. These studies showed that mutation of T688 to an alanine increases total kinase autophosphorylation (ie autoactivity) and the subsequent phosphorylation of DCX domains, which in turn decreases microtubule binding. Given this information, our goal was to use an evolutionary perspective to investigate this, alongside less-well characterized aspects of DCLK autoregulation, including co-conserved residues in the catalytic domain and C-terminal tail. However, to address the reviewers question of a non-phosphorylatable T688 mutant, we performed MD simulations of T688A and T688E (a phosphomimic) mutant and include a new supplementary figure (Figure 5-supplement 3) which show the two mutants slightly destabilize the C-tail relative to wt (1 and 2 angstrom increase in RMSF for T688E and T688A respectively), but by themselves cannot dislodge the C-tail from the ATP binding pocket. Thus, other co-conserved interactions as revealed by our analysis, are likely to contribute to the autoregulation of the kinase domain by the C-terminal tail. We have incorporated these observations into the revised results section.

Furthermore, to address the reviewer’s question in terms of site-specific autophosphorylation as a marker of DCLK1.2 activity, we have now performed a much-more detailed phosphoproteomic analysis of a panel of purified DCLK1.2 proteins after purification from *E. coli* (Figure 8-figure supplement 2). This showed that we are only able to detect Thr 688 phosphorylated in our ‘activated’ DCLK1.2 mutants, and not in the autoinhibited WT DCLK1.2 version of the protein. This apparent contradiction does not necessary discount Thr 688 as an important regulatory hotspot, but, together with the MD simulations, may imply a decreased contribution of pThr 688 in facilitating/maintaining DCLK1.2 auto-inhibition than previously anticipated, especially in the context of the numerous other stabilizing amino acid contacts that we describe between the C-tail and the ATP-binding pocket. We do, however, propose a mechanism for pThr688 as a potential ATP mimic based on MD analysis. However, we only found MS-based evidence for phosphorylation at this (and other sites in the same peptide) in highly active DCLK1.2 mutants, in which the C-tail remains uncoupled from the ATP-binding site, even in the presence of this regulatory PTM. We acknowledge that better understanding of DCLK biology will require a detailed appraisal of how the DCLK auto-inhibited states are subsequently physiologically regulated (PTMs, protein-protein interaction etc.), but this is beyond the scope of our current evolutionary investigation, and the absence of phosphospecific antibodies makes this challenging currently. We intend to expand upon our current work by assessing the relative contribution of multiple DCLK phosphorylation sites (including, but not limited to, Thr 688) with regard to cellular DCLK auto-regulation in future studies, in part by generating such site-specific phospho-antibodies.

1. Have the authors made an equivalent T687/688 tanden in DCLK1.1 instead of the two prolines?

This is a good point. We have not considered introducing a T687/688 tandem mutation into DCLK1.1 (at the equivalent position to that of DCLK1.2), primarily because the amino acid composition of their respective C-tail domains are so highly divergent across the tail (due to alternative splicing, as discussed in our paper). As discussed in our present study, there are numerous contacts made between specific amino acids in the regulatory C-tail and the kinase domain of DCLK1.2, which functionally occlude ATP binding, and thus change catalytic output. It is these contacts, which are determined by the specific amino acid sequence identity, and not the extended length of the DCLK1.2 C-tail per se, that drives autoinhibition. The alternate amino acid sequence identity of the C-tail of DCLK1.1 does not enable such contacts to form, which we believe explains the different activities of the two isoforms.

Furthermore, our mutational analysis reveals clearly that Thr688 and several other sites are more highly autophosphorylated in the artificially activated DCLK1.2 constructs than WT DCLK1.2, and as such it remains our hypothesis that introduction of the tandem phosphorylation sites into DCLK1.1 is unlikely to be sufficient to impose an auto-inhibitory conformation of the enzyme.

1. Could T688 autophosphorylation be used as a functional readout to evaluate DCLK1.2 activity?

We agree with the reviewer’s suggestion about using autophosphorylation (including potentially Thr688 for DCLK1.2) as a functional read out for DCLK1 activity. In our present study, we identify phosphorylated peptides containing pThr688 only in the mutationally activated DCLK1.2 variants. We have now taken this analytical approach further and performed a detailed comparative phosphoproteomic characterisation of all of our DCLK1 constructs, where we observe marked differences in the overall phosphorylation profiles of the mutant DCLK1.2 (and DCLK1.1) proteins relative to the less phosphorylated WT DCLK1.2 kinase. This manifests as a depletion in the total number of confidently assigned phosphorylation sites within the kinase domain and C-tail of WT DCLK1.2, and also as a depletion in the abundance of phosphorylated peptides for a given site. To help visualise this, individual phosphorylation sites have been schematically mapped onto DCLK1, which has been included as a new extended supplementary figure (Figure 8-figure supplement 2). For comparative analysis of phosphosite abundance, we could only select peptides that could be directly compared between all mutants (identical amino acid sequences) and those found to be phosphorylated in all proteins (these are Ser660 and Thr438); these are now shown in figure supplement 2 as a table. These site occupancies follow what we see with respect to the increased catalytic activity between DCLK1.1 and DCLK1.2 mutants versus DCLK1.2. We also detect increased phosphorylation of DCLK1.1 and activated DCLK1.2 mutants in comparison to (autoinhibited) DCLK1.2, supporting the hypothesis that these mutants are relieving the autoinhibited conformation.

1. What are the evidences of the here described c-terminal specific interactions to be intra-molecular rather than inter-molecular? Have the authors looked at the monodispersion and molecular mass in solution of the different protein evaluated in this study? Basically, are the proteins in solutions monomers or dimers/oligomers?

Analysis of symmetry mates in the crystal structure of DCLK1.2 (PDB ID: 6KYQ) provide no evidence for inter-molecular interactions. Furthermore, to evaluate oligomerization status in solution, we conducted an analytical size exclusion chromatography (SEC) and our analysis reveals that both DCLK1.1 and DCLK1.2 predominantly exist as monomers in solution (Figure 3-Supplements 1-3). These results suggest that the C-terminal tail interactions are primarily intra-molecular.

1. (Figure 3) Did the authors look at the mono-dispersion of the protein preparation? The sec profile did result in one single peak or multiple peaks? Could the authors show the chromatogram? how many species do you have in solution? Was the tag removed from the recombinant proteins or not?

Yes, as mentioned above, the SEC profile resulted in a single peak for both DCLK1.1 and DCLK1.2, which was confirmed as DCLK1 by subsequent SDS-PAGE. We have included the chromatogram and gels in supporting data (Figure 3-supplements 1-3) in the revised manuscript and updated the Methods section. ‘The short N-terminal 6-His affinity tag present on all other DCLK1 proteins described in this paper was left in situ on recombinant proteins, since it does not appear to interfere with DSF, biochemical interactions or catalysis.’

1. Authors should do Michaelis-Menten saturation kinetics as shown in Figure 3C with the WT when comparing all the functional variant analysed in the study. So we can compared the catalytic rates and enzymatic constants (depicted in a table also) kcat, Km and catalytic efficiency constants (kcat/Km)

Thank you for your suggestion. We have performed the requested comparative kinetics analyses for selected functional DCLK1 variants at the same concentration as suggested, using our real-time assay to determine Vmax for peptide phosphorylation as a function of ATP, but at a fixed substrate concentration (we are unable to assess Vmax above 5 µM peptide for technical reasons). The results of these analyses have been included in the revised version of Figure 8-Supplement 1, where they support differences in both Vmax and Km[ATP]; the ratio of these values very clearly points to differences in activities falling into ‘low’ or ‘high’. This kinetic analysis fully supports our initial activity assays, where mutations predicted to uncouple the auto-inhibitory C-tail rescue DCLK1.2 activity to levels similar to DCLK1.1 towards a common substrate.

Minor CommentsIt is very interesting how the IBS together with the pT688 mimics ATP in the case of DCLK1.2 to reach full occupancy of the active site. On Figure 8 you evaluate residues of the GRL and IBS interface to probe such interactions.1. Did the authors look at the T688 non-phosphorylable mutant?

See our response to Major Comment 1 above. In addition, due to the absence of T688 in DCLK1.1, we did not look at the T688A mutant of DCLK1.2 biochemically, partially because it has been characterized in previous studies, but partially because this site is preceeded by another Thr residue. The lack of a selective antibody towards this site makes it difficult to evaluate the role of T688 phosphorylation specifically with respect to DCLK cellular functions and interactions. Therefore, we focused our in vitro efforts to understand how mutations in the IBS impact the catalytic activity of DCLK1.2 by comparing different variants to DCLK1.1.

1. Classification of DCLK C-terminal regulatory elements.It would be useful to connect the different regulatory elements described in this study to a specific functional and biological setting where these different switches play a role e.g. microtubule interactions and dynamics, cell cycle, cancer, etc..

While the primary focus of our paper is on the mechanism of allosteric regulation of DCLK1, we have indeed touched upon the potential implications of the various regulatory elements of the tail on functions such as microtubule binding and phenotypic effects like cancer progression. However, we acknowledge that a comprehensive understanding of these effects would necessitate a more detailed investigation. This could potentially involve the integration of RNA-seq data with extensive cell assays to evaluate phenotypic effects. We believe that such a future study would be a valuable extension of our current work and could provide further insights into the functional roles of DCLK1.

1. (Figure 3) Could the authors explain the differences in yield between the WT and the D531A mutant. Apparently, it [the yield] does not appear to be caused by a lower stability as indicated by the Tm. Could the authors comment on this?It is important to compare different samples in parallel, in the same experiment and side by side. This applies to the thermal shift data comparing WT and a D531A mutant on panel D and also on panel C a comparison between WT and D531A as negative control should be shown.

WT and D533A (kinase-dead) were indeed analysed in parallel, but have been split in two panels to make the data easier to interpret. The modest differences in yield is likely explained by experimental prep-to-prep variations. Our experience shows that many protein kinase yields vary between kinase and kinase-dead variants, likely due to bacterial toxicity related to enzyme activity. In regards to thermal stability, we would like to emphasize that Differential Scanning Fluorimetry (DSF) is to our mind a more informative and quantitative measure of protein stability than yield from bacteria, because both assess purified proteins at the same concentration. We believe that the DSF data provide a more accurate representation of the real stability differences between the WT and D533A mutant.